# MedAgent-Pro: Towards Evidence-based Multi-modal Medical Diagnosis via Reasoning Agentic Workflow

**Ziyue Wang**[1], **Junde Wu**[2], **Linghan Cai**[3], **Chang Han Low**[1],
**Xihong Yang**[1,4], **Qiaxuan Li**[5], **Yueming Jin**[1]*
[1]National University of Singapore, [2]University of Oxford,
[3]Harbin Institute of Technology (Shenzhen), [4]National University of Defense Technology,
[5]The Second Affiliated Hospital Zhejiang University School of Medicine

## Abstract

Modern clinical diagnosis relies on the comprehensive analysis of multi-modal patient data, drawing on medical expertise to ensure systematic and rigorous reasoning. Recent advances in Vision–Language Models (VLMs) and agent-based methods are reshaping medical diagnosis by effectively integrating multi-modal information. However, they often output direct answers and empirical-driven conclusions without clinical evidence supported by quantitative analysis, which compromises their reliability and hinders clinical usability. Here we propose MedAgent-Pro, an agentic reasoning paradigm that mirrors modern diagnosis principles via a hierarchical diagnostic workflow, consisting of disease-level standardized plan generation and patient-level personalized step-by-step reasoning. To support disease-level planning, a retrieval-augmented generation agent is designed to access medical guidelines for alignment with clinical standards. For patient-level reasoning, MedAgent-Pro leverages professional tools such as visual models to take various actions to analyze multi-modal input, and performs evidence-based reflection to iteratively adjust memory, enforcing rigorous reasoning throughout the process. Extensive experiments across a wide range of anatomical regions, imaging modalities, and diseases demonstrate the superiority of MedAgent-Pro over mainstream VLMs, agentic systems and leading expert models. Ablation studies and expert evaluation further confirm its robustness and clinical relevance. Anonymized code link is available in the reproducibility statement.

## 1 Introduction

As a core task in medical practice, clinical diagnosis entails synthesizing various clinical information to reach a conclusion (Steinberg et al., 2011; Guyatt et al., 1992a), where clinicians make decisions mainly based on visual cues from medical imaging and textual information from patient records. For example, it is common for clinicians to examine radiology images to identify tumor scales, or analyze pathological slides to detect potential cancer. Early AI-assisted methods focus on adopting pure image analysis models to support diagnosis, such as pneumonia severity assessment in chest radiology or cellular classification in pathology. Recently, Vision-Language Models (VLMs) have demonstrated that integrating multi-modal information can significantly benefit diagnosis, and medical visual question answering (VQA), where models answer textual questions based on images, has become a key benchmark in this context (Zhan et al., 2020; Chen et al., 2025b; Liang et al., 2024).

However, ordinary VQA formulated in existing methods fails to reflect the real-world diagnostic processes. Rather than performing one-hop VQA, clinical diagnosis involves a standardized, step-by-step process (Steinberg et al., 2011; Eddy, 1990a;b;c; Albarqouni et al., 2018). The diagnostic process for each patient typically involves two stages: i) Determining the target disease and formulating a standardized workflow based on medical guidelines with clinical indicators to support the decision; ii) Step-by-step analysis of personalized data, combining qualitative and quantitative

---

*Corresponding author: Yueming Jin (ymjin@nus.edu.sg)

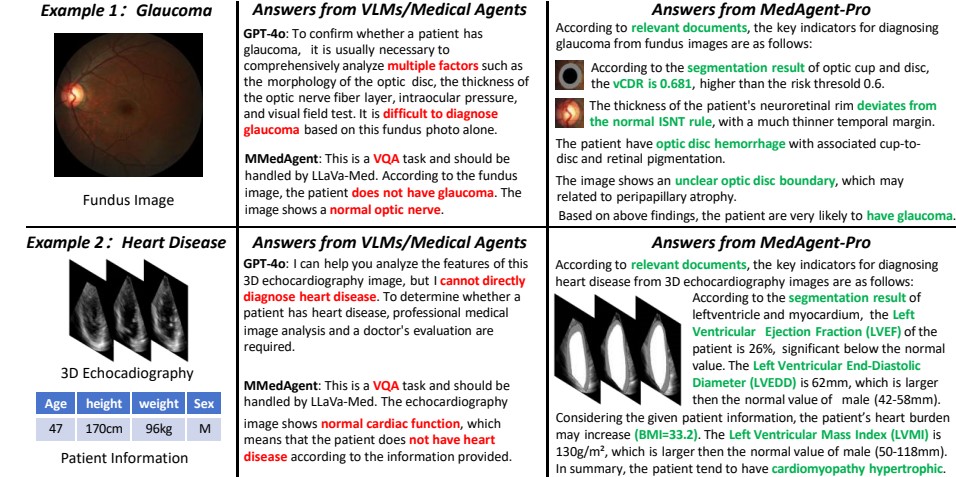

Figure 1: Comparison of diagnostic outcomes for two diseases across mainstream VLMs, medical agentic systems, and our proposed MedAgent-Pro workflow.

assessments to evaluate these indicators. Throughout the process, each step builds on previous assumptions and is supported by relevant literature or specialized tools For example, in glaucoma diagnosis, the cup-to-disc ratio is a key indicator, which relies on the accurate localization of the optic cup and disc in the preceding step. In contrast, current VQA models often generate diagnostic conclusions hastily, relying on empirical internal knowledge without fine-grained analysis.

Due to the dual requirements of patient safety and evidence traceability mandated by clinical regulation, current methods fall short of meeting the standard: VLMs (Achiam et al., 2023; Chen et al., 2025a) have shown strong performance across a range of tasks (Ghezloo et al., 2025; Benary et al., 2023). However, they lack sufficient medical knowledge and remain inadequate for in-depth medical analysis. While models like GPT-o1 (Achiam et al., 2023) and DeepSeek-R1 (Guo et al., 2025) incorporate reasoning capabilities to support more structured analysis, their limited fine-grained visual perception ability impairs quantitative analysis and reduces their effectiveness in clinical applications. Meanwhile, agentic systems (Chen et al., 2023; Li et al., 2023) have extended the capabilities of VLMs by introducing professional tools, but current medical agentic systems (Kim et al., 2024; Zuo et al., 2024; Fallahpour et al., 2025) integrate tools statically instead of dynamically organizing tools according to the given disease. As a result, when asked to provide a diagnosis, these systems simply conduct a fixed pipeline without selecting and orchestrating appropriate tools to support their decision-making. In summary, existing methods treat medical diagnosis as an empirical one-hop QA task, drawing exclusively from VLMs' internal knowledge to make qualitative judgments. In contrast, medical practice emphasizes evidence-based diagnosis, requiring structured reasoning and clinical indicators (Eddy, 1990a;b;c; Zhu et al., 2025a).

To tackle these issues, we present MedAgent-Pro, a reasoning agentic workflow tackling versatile multi-modal medical diagnosis tasks. We aim to design a workflow that aligns with modern medical criteria, provides decision support with medical guidelines and quantitative analysis as shown in Fig. 1. Our MedAgent-Pro embraces a hierarchical structure to simulate the modern clinical procedure, conduct step-by-step reasoning at the disease and patient levels. *Disease-level planning* generates standardized diagnostic plans, while *Patient-level reasoning* follows these plans to analyze personalized information. To facilitate alignment with clinical guidelines, MedAgent-Pro incorporates a retrieval-augmented generation (RAG) agent that injects relevant medical knowledge into the diagnostic planning. In the reasoning process, MedAgent-Pro integrates expert tools such as visual models to take various actions for diagnosis support, enabling accurate qualitative and quantitative analysis of clinical indicators. Furthermore, to maintain the rigor of multi-step clinical reasoning, we propose an evidence-based reflection mechanism, whereby the system evaluates the reliability of each step's output and adjusts the memory to guide subsequent actions, ensuring that every decision is grounded in a sound and trustworthy foundation. Our contributions are summarized as:

- We propose MedAgent-Pro, the first agentic paradigm that presents systematic, evidence-based reasoning for accurate and reliable medical diagnosis. By aligning with the principle of modern

medical workflow, our paradigm transforms the empirical, ready-made outputs of prior methods into more rigorous logical reasoning and a well-founded conclusion.

- MedAgent-Pro presents a hierarchical structure consisting of *disease-level* and *patient-level* reasoning. A RAG-based method makes the disease-level planning align with medical guidelines, while tool-based action and evidence-based reflection are devised at the patient level to ensure professionalism and reliability of the step-by-step analysis.

- Evaluations across 10+ imaging modalities, 20+ anatomies, and 50+ diseases show that MedAgent-Pro comprehensively surpasses mainstream VLMs, agentic systems, and task-specific models. Notably, it outperforms GPT-4o by 34% and 22% on glaucoma and heart disease diagnosis. Clinician evaluations further highlight the diagnostic quality and reliability of our method.

## 2 RELATED WORK

**Multi-modal Medical Diagnosis.** Developing AI techniques for multi-modal medical diagnosis has become a primary research objective (Bakator & Radosav, 2018; Kononenko, 2001; McPhee et al., 2010). Prior research focused on medical imaging assessment, including classification (Zhang et al., 2023a; 2019; Azizi et al., 2021; Association, 2020), detection (Baumgartner et al., 2021; Wang et al., 2020; Bejnordi et al., 2017), and segmentation (Isensee et al., 2021; Wu et al., 2025; Ma et al., 2024a). VQA has been proposed for end-to-end multi-modal diagnosis (Zhang et al., 2023b; Khare et al., 2021; Zhang et al., 2025), while VLMs (Li et al., 2024b; Moor et al., 2023; Liang et al., 2025; Lin et al., 2024) have yielded competitive performance in medical VQA. Despite these advancements, medical VQA (Lau et al., 2018; Liu et al., 2021; He et al., 2020) remain overly simplified compared to the diagnostic practice, and further research is highly desired.

**VLM-based AI Agent.** Establishing an autonomous intelligent system is a long-standing research goal, with agent-based methods gaining increasing attention. VLM-based agents have made great progress in diverse applications such as industrial engineering (Mehta et al., 2023; Xia et al., 2023), scientific experimentation (Boiko et al., 2023), embodied agents (Brohan et al., 2023; Huang et al., 2022), gaming (Gallotta et al., 2024), and societal simulation (Ma et al., 2024b; Jinxin et al., 2023). Despite their adaptability in various scenarios, VLM-based agents remain limited in the medical domain due to insufficient fine-grained visual perception.

**Medical Agentic System.** Current medical agentic systems can be categorized into two streams. The first stream enhances VLM capabilities through mechanisms such as debate or majority voting among agents to refine responses (Kim et al., 2024; Tang et al., 2024; Zuo et al., 2024; Lyu et al., 2025; Zhu et al., 2025b) . The second ones integrate an orchestrator agent with various specialized models to solve various medical tasks (Li et al., 2024a; Xia et al., 2024; Fallahpour et al., 2025; Fathi et al., 2025; Wang et al., 2026). Nevertheless, they mainly aggregate tools in a fixed manner instead of a clinically-grounded workflow, leading to suboptimal handling of complex diagnoses.

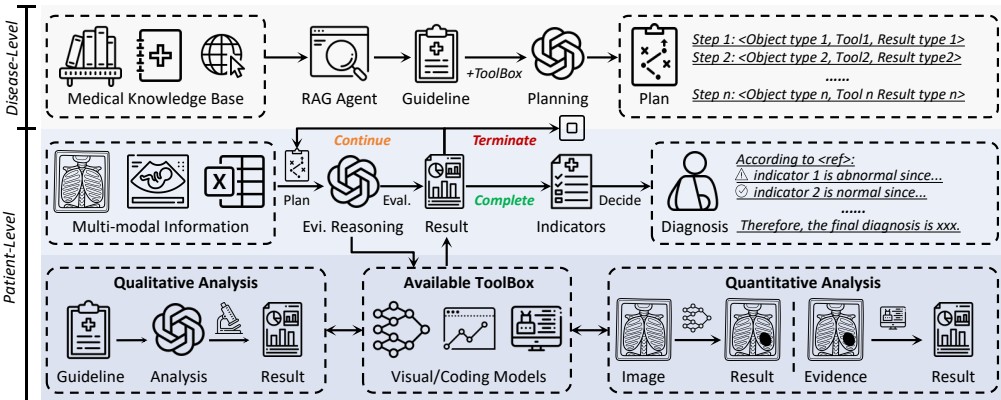

Figure 2: Overview of the MedAgent-Pro. MedAgent-Pro performs diagnosis through a hierarchical structure, with reasoning guided by a VLM supported by an RAG agent and specialized tools. Evi. means Evidence-based in the figure.

## 3 METHODS

### 3.1 OVERALL WORKFLOW

For a specific disease, clinicians typically follow guidelines to develop a standardized diagnostic workflow. Each patient's diagnosis then follows this workflow, combining qualitative observations and quantitative assessments to obtain clinical indicators (Details in Appendix D). Emulating this paradigm, MedAgent-Pro design a hierarchical reasoning workflow consisting of *disease-level planning* and *patient-level reasoning*, enabling personalized diagnosis under standardized, disease-specific guidance. We take a VLM $\mathcal{V}$ as an orchestrator to conduct basic tasks within the workflow.

As illustrated in Fig. 2, the *disease-level planning* is designed to formulate standardized diagnostic plans for each disease based on medical guidelines, which is assisted by an RAG agent. In the meantime, *patient-level reasoning* processes each patient's personalized data individually by executing the diagnostic plan step by step and assisting the VLM in performing necessary quantitative analysis through specialized tools, thereby enabling evidence-based reasoning as follows:

### 3.2 DISEASE-LEVEL KNOWLEDGE-BASED PLANNING

Doctors typically develop standardized workflows based on their expertise and medical guidelines (Eddy, 1990b;c; Shaneyfelt et al., 2006). Aligning with this practice, we introduce a Retrieval-Augmented Generation (RAG) agent $\mathcal{R}$ to incorporate medical guidelines during the planning stage to guide diagnostic plan generation. MedAgent-Pro is equipped with a large-scale knowledge base $\mathcal{K}$, built from MedlinePlus (Miller et al., 2000), which includes entries on 1,000+ diseases and conditions, and 4,000+ expert-reviewed corresponding guidelines authenticated by NIH/NLM.

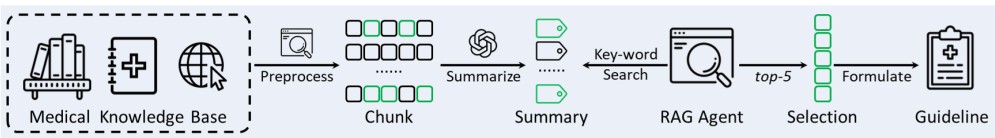

Figure 3: The illustration of the RAG process, which leverages a two-step retrieval.

As shown in Fig. 3, we first index $\mathcal{K}$ to support efficient retrieval. Each guideline article is first assigned a short summary of its metadata like organ and disease in MedlinePlus. Given a disease query, the RAG agent $\mathcal{R}$ first filters documents using the summary to obtain a query-relevant subset. The full article is then segmented into 300-token chunks, embedded with PubMedBERT (Gu et al., 2021), and stored in a vector index. $\mathcal{R}$ then retrieve the top-5 most relevant chunks based on similarity. Based on the retrieved chunks, the VLM summarizes their content and generates a procedural guideline $\mathcal{G} = \mathcal{V}(\mathcal{R}(\mathcal{K}))$ that reflects real-world clinical practices for the queried disease.

For the diagnostic plan generation, the VLM summarizes disease-specific clinical indicators $\mathcal{I} = \{\mathcal{I}_1, \mathcal{I}_2, \ldots, \mathcal{I}_m\}$ from the guideline $\mathcal{G}$. To support the analysis of $\mathcal{I}$, MedAgent-Pro is equipped with a toolset $\mathcal{T}$ and an associated action set $\mathcal{A}$, where each action $a \in \mathcal{A}$ corresponds to a tool $t \in \mathcal{T}$ through a predefined mapping $\psi(a) = t$. For example, $t$ may be a segmentation model, and $a$ is its paired action such as "Segments the optic cup in a fundus image." Based on $\mathcal{G}$ and $\mathcal{A}$, the VLM generates a disease-specific diagnostic plan $\mathcal{P} = \mathcal{P}1, \mathcal{P}2, \ldots, \mathcal{P}n$. Each step $P_i$ applies an action $a_i \in \mathcal{A}$ to an input object $o_i$ and produces an output $r_i$:

$$P_i : r_i = a_i(r_i), a_i \in \mathcal{A}, \tag{1}$$

Here $o_i$ and $r_i$ can be the input data like fundus image, intermediate result like segmentation mask or final indicators. In practice, $\mathcal{P}$ is stored as a JSON file. Each $\mathcal{P}_i$ includes an operation action $a_i$, a predefined Python function from the toolset $\mathcal{T}$ with fixed behavior, and two data fields specifying the expected input and output data property for $a_i$. Throughout the design, each disease is assigned a diagnosis plan aligned with medical guidelines, enabling a regulated and standardized workflow.

### 3.3 PATIENT-LEVEL EVIDENCE-BASED REASONING

Grounded in the evidence-based principle of modern medicine, which integrates reliable clinical evidence with individual expertise to guide patient care, MedAgent-Pro follows the disease-specific diagnostic plan $\mathcal{P}$ to analyze clinical indicators $\mathcal{I}$ for each patient case in the following paradigm:

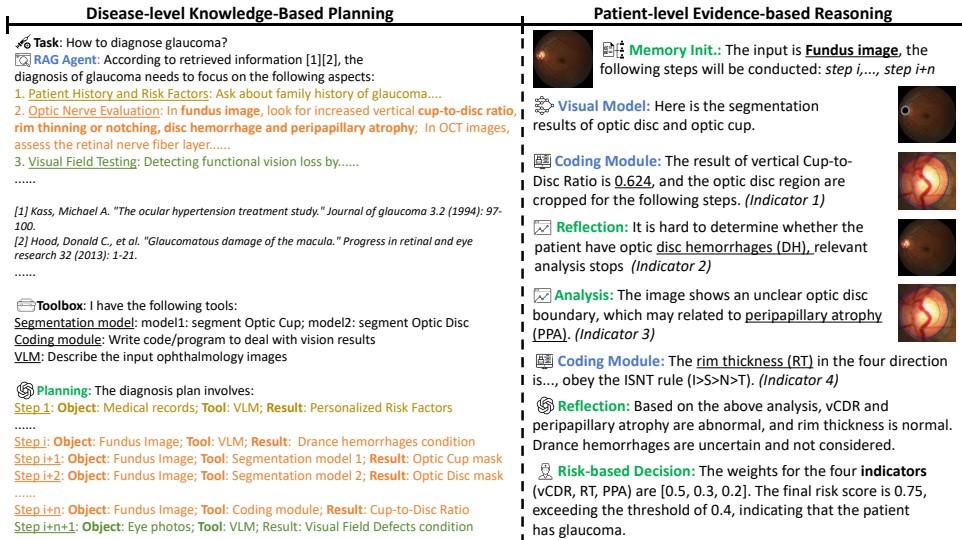

Figure 4: A detailed case study for glaucoma diagnosis. The blue text indicates the agents, while the green text indicates the reasoning steps. More cases can be found in Appendix F.

**Memory Initialization.** For each patient's personalized multi-modal data $\mathcal{D}$, the VLM performs an orchestration process to select executable steps from $\mathcal{P}$ based on data availability, filtering out those requiring unavailable inputs to form a long-term memory $\mathcal{M}$ for each patient's diagnosis:

$$\mathcal{M} = \{\, P_i \in \mathcal{P} \mid o_i \in \mathcal{D} \,\}. \tag{2}$$

As Fig. 4 shows, in the glaucoma diagnosis, when provided with the fundus image, the orchestration process selects relevant steps and skips those requiring unavailable data such as OCT scans.

**Tool-based Action.** During the reasoning, the VLM checks the data property of the current input: if it matches any $o_i$, MedAgent-Pro conducts corresponding step $P_i$ stored in memory $M$ to obtain $r_i$. For $a_i$, while qualitative analysis is conducted directly by the VLM, quantitative analysis relies on professional tool agents in the toolset to perform specialized assessments. $\mathcal{T}$ includes visual models such as segmentation tools (e.g., MedSAM (Ma et al., 2024a), Cellpose (Stringer & Pachitariu, 2025)) and LLM-based coding tools (e.g., Copilot (github team)), where thorough descriptions and ablation studies of toolset are in Appendix B. As shown in Fig. 4, specialized segmentation tools are used to extract the optic cup and disc masks, while coding tools subsequently compute the cup-to-disc ratio based on the segmentation results, which serves as a key indicator in glaucoma diagnosis.

**Evidence-based Reflection.** During sequential reasoning, the system evaluates the output $r_i$ at each step $\mathcal{P}'_i$ to determine a status $s_i \in \{Continue, Terminate, Complete\}$ as short-term memory, which indicates how MedAgent-Pro take the subsequent action. $s_i$ is determined by:

$$s_i = \begin{cases} Complete & r_i \in \mathcal{I}, \\ Terminate & r_i \notin \mathcal{I} \ \wedge \ \neg\phi(r_i, o_i, G), \\ Continue & r_i \notin \mathcal{I} \ \wedge \ \phi(r_i, o_i, G). \end{cases} \tag{3}$$

Here, $\phi$ is a state assessment function implemented by the VLM, which evaluates the reliability of $r_i$ based on the quality of input data $o_i$, and the plausibility of the result (Details in Appendix E.4.) If $s_i = Continue$, the output $r_i$ is treated as evidence $e$ and used as the input $o_{i+1}$ for the next reasoning step. If $s_i = Terminate$, the process is halted, as $r_i$ is deemed unreliable and may hinder subsequent reasoning and lead to incorrect diagnoses. This process continues iteratively until $s_i = Complete$.

**Risk-based Decision.** The reasoning results in a set of indicators: $\mathcal{R}_{final} = \{r_i | s_i = Complete\}$. The VLM then assigns risk-based weights $\mathcal{W}$ for $\mathcal{R}_{final}$ according to clinical importance grounded in medical guideline: $W = \mathcal{V}(\mathcal{R}_{final}|G)$. The final risk score $\rho$ is computed as a weighted sum:

$$\rho = \sum_{i=0}^{|\mathcal{R}_{final}|=l} w_i r_i, s.t. w_i \in W, r_i \in \mathcal{R}_{final} \,. \tag{4}$$

| Method | Glaucoma | | Heart Disease | | NEJM (Acc) | | | | |
|---|---|---|---|---|---|---|---|---|---|
| | bAcc | F1 | bAcc | F1 | All | Cell Imaging | Chest X-ray | CT & MRI | Opht. Imaging |
| *General VLMs* | | | | | | | | | |
| GPT-4o | 56.4 | 21.1 | 56.8 | 28.1 | 70.9 | 74.6 | 54.5 | 63.4 | 70.5 |
| Janus-Pro-7B | 53.4 | 13.3 | 52.3 | 10.7 | 30.0 | 32.2 | 36.4 | 29.3 | 28.2 |
| LLaVA-Med | 50.0 | 0.0 | 50.0 | 0.0 | 26.2 | 39.0 | 18.2 | 24.4 | 35.9 |
| BioMedClip | 58.1 | 21.3 | 47.0 | 37.8 | 27.9 | 27.1 | 40.0 | 29.3 | 20.5 |
| Qwen2.5-7B-VL | 54.3 | 16.3 | 50.0 | 0.0 | 41.8 | 54.2 | 30.9 | 36.6 | 44.9 |
| InternVL2.5-8B | 51.8 | 13.8 | 49.7 | 3.6 | 42.2 | 37.3 | 40.0 | 41.5 | 35.9 |
| *Medical Agentic Systems* | | | | | | | | | |
| MedAgents (ACL'24) | 52.1 | 8.9 | 51.1 | 15.9 | 66.1 | 69.9 | 51.5 | 58.7 | 68.2 |
| MMedAgent (EMNLP'24) | 52.4 | 16.3 | 55.0 | 26.7 | 71.7 | 73.8 | 56.4 | 65.3 | 70.5 |
| MDAgent (NeurIPS'24) | 56.8 | 22.2 | 57.2 | 30.3 | 73.8 | 79.6 | 52.9 | 67.3 | 73.0 |
| MedAgent-Pro (Ours) | **90.4** | **76.4** | **77.8** | **72.3** | **81.7** | **90.5** | **69.1** | **72.7** | **89.7** |

Table 1: Comparison with general VLMs and medical agentic systems on REFUGE2, MITEA and NEJM (%). "Opht." is the short form of Ophthalmology. Our setting is highlighted in green .

| Method | Avg. | Atelectasis | Cardiomegaly | Consolidation | Edema | Enlarged Cardiomediastinum | Fracture |
|---|---|---|---|---|---|---|---|
| GPT-4o | 58.3 | 68.7 | 64.3 | 60.5 | **61.2** | 53.2 | 56.3 |
| Janus-Pro-7B | 51.9 | 61.7 | 54.1 | 45.2 | 52.5 | 34.1 | 50.0 |
| LLaVA-Med | 50.2 | 50.0 | 51.5 | 50.0 | 50.0 | 50.0 | 50.0 |
| BioMedClip | 57.9 | 48.6 | 62.7 | 62.2 | 50.0 | 61.1 | 50.0 |
| Qwen2.5-7B-VL | 55.2 | 69.6 | 58.1 | 54.5 | 48.8 | 50.8 | 50.0 |
| InternVL2.5-8B | 51.6 | 57.7 | 48.6 | 51.7 | 47.5 | 50.8 | 43.8 |
| MedAgent-Pro | **72.0** | **85.5** | **74.2** | **66.3** | 59.2 | **75.4** | **68.5** |

| Method | Avg. | Lung Leision | Lung Opacity | Pleural Effusion | Pneumonia | Pneumothorax | Supporting Devices |
|---|---|---|---|---|---|---|---|
| GPT-4o | 58.3 | 38.6 | 63.1 | 62.7 | 59.6 | 47.7 | 63.4 |
| Janus-Pro-7B | 51.9 | 59.1 | 42.1 | 57.4 | 57.4 | 47.2 | 62.4 |
| LLaVA-Med | 50.2 | 50.0 | 50.0 | 50.0 | 50.1 | 50.5 | 50.0 |
| BioMedClip | 57.9 | **81.8** | 50.0 | 64.7 | 55.1 | 53.0 | 55.4 |
| Qwen2.5-7B-VL | 55.2 | 43.2 | 64.4 | 48.2 | 59.3 | 34.9 | 80.8 |
| InternVL2.5-8B | 51.6 | 59.1 | 49.1 | 54.2 | 54.0 | 47.6 | 55.3 |
| MedAgent-Pro | **72.0** | 72.2 | **67.6** | **77.8** | **62.1** | **65.6** | **89.2** |

Table 2: Comparison with general VLMs on the MIMIC dataset (%). "Avg." is the average performance across 12 sub-tasks. Only bAcc values are presented; F1 score can be found in Appendix A.

The final diagnosis is then made by comparing $s$ with a risk threshold $\theta$. Through evidence-based reasoning, MedAgent-Pro integrates reliable external evidence with expert knowledge to improve diagnostic decision-making, thus promoting the modern diagnosis workflow.

# 4 EXPERIMENT

## 4.1 EXPERIMENTAL SETUP

**Dataset.** We conduct experiments on four datasets. The REFUGE2 dataset (Fang et al., 2022) is used for glaucoma diagnosis, and the MITEA dataset (Zhao et al., 2023) for heart disease diagnosis, both of which require in-depth diagnostic reasoning. To assess multi-disease diagnosis for individual patients, we sample 442 chest X-ray cases from 100 patients from the MIMIC dataset (Johnson et al., 2023), each case involving the identification of up to 12 potential thoracic abnormalities. We further employ the New England Journal of Medicine (NEJM) database (of Medicine), where we compile 992 real-world diagnostic cases, encompassing 10+ anatomical regions, 10+ imaging modalities, and 50+ diseases. For cases involving cell or ophthalmology imaging, visual tools are available. However, non-clinical images (e.g., daily photographs) often lack compatible tool support.

**Evaluation Metrics.** Three metrics are used to evaluate performance. For REFUGE2, MITEA, and MIMIC datasets, we report balanced accuracy (bAcc) and the F1 score. For the NEJM dataset, where tasks are framed as multiple-choice questions in accordance with its evaluation protocol, we report the accuracy rate. The best results are highlighted in **bold**, and the second-best are underlined.

| REFUGE2 winners | | Ophthalmology VLMs | | | Chest X-Ray VLMs | |
|---|---|---|---|---|---|---|
| Team Name | AUC | Method | bAcc | F1 | Method | bAcc |
| VUNO EYE TEAM | 88.3 | RetiZero | 50.8 | 18.4 | Maira-2 | 64.1 |
| MIG | 87.6 | VisionUnite | 85.8 | 73.1 | CheXagent | 69.1 |
| MedAgent-Pro | **95.1** | MedAgent-Pro | **90.4** | **76.4** | MedAgent-Pro | **72.0** |

Table 3: Comparison with task-specific models on diverse datasets (%).

**Implementation Details.** In the MedAgent-Pro, we use GPT-4o (Achiam et al., 2023) as the baseline VLM by default, and implement the RAG agent using LangChain (Topsakal & Akinci, 2023). For fair comparison, all baseline medical agentic systems adopt GPT-4o as their underlying VLM.

## 4.2 PERFORMANCE EVALUATIONS

**Comparison with General VLMs.** We compared with advanced VLMs including BioMedClip (Zhang et al., 2023a), GPT-4o (Achiam et al., 2023), LLaVA-Med (Li et al., 2024b), Janus-Pro (Chen et al., 2025a), Qwen (Wang et al., 2024b), and InternVL (Chen et al., 2024a). As these VLMs cannot handle 3D images, we sample three random slices from each 3D echocardiography in the MITEA dataset, repeat this ten times, and report the mean performance to reduce sampling variability (std. value in Appendix A). As shown in Table 1 and 2, the proposed MedAgent-Pro framework significantly outperforms existing VLMs across all datasets. Compared to GPT-4o, it achieves improvements in bAcc of 34.0% and 21.0%, and gains in F1 score of 55.3% and 44.2% for glaucoma and heart disease diagnosis respectively. This highlights MedAgent-Pro workflow's effectiveness in handling complex diagnostic tasks, particularly those requiring quantitative indicators such as cup-to-disc ratio or left ventricular ejection fraction. Integration of visual tools into MedAgent-Pro's reasoning enables precise indicator calculation and diagnosis, addressing current limitations of VLMs. More comparison results and detailed analyses are in Appendix A.

Across diverse real-world diagnostic scenarios in the NEJM (details in Appendix E), MedAgent-Pro demonstrates significant performance gains in domains with visual tool support, such as cell imaging, chest X-rays, CT/MRI, and ophthalmology. Furthermore, it even maintains strong performance in cases without visual tool support, achieving an overall improvement of 7.9%, which demonstrates its strong robustness and generalizability across a variety of diagnostic tasks. In addition, as shown in Table 2, chest X-ray diagnosis involves certain tasks like *Cardiomegaly* which rely on precise quantitative measurements like the cardiothoracic ratio. Meanwhile, others such as *Fracture* detection, require detailed step-by-step analysis to identify subtle abnormalities. MedAgent-Pro achieves leading performance across most tasks, with an average performance gain of 13.7%.

**Comparison with Medical Agentic Systems.** We also compare MedAgent-Pro with advanced medical agentic frameworks, including MedAgents (Tang et al., 2024), MMedAgent (Li et al., 2024a) and MDAgent (Kim et al., 2024). As shown in Table 1, MedAgent-Pro consistently outperforms these methods across all diseases and domains. This performance advantage stems from the fact that prior methods are primarily designed for basic question answering or as modular toolboxes, lacking the capacity to handle complex, multi-modal clinical scenarios. In contrast, MedAgent-Pro incorporates retrieval-based diagnostic steps and seamlessly integrates visual tools into its reasoning process, enabling effective and comprehensive decision-making support in clinical applications.

**Comparison with Task-Specific Models.** Additionally, we compare MedAgent-Pro with state-of-the-art task-specific methods, i.e., for glaucoma diagnosis (Wang et al., 2024a; Li et al., 2024c) and chest X-ray analysis (Bannur et al., 2024; Chen et al., 2024b). For glaucoma diagnosis, we also compare with the winners from the REFUGE2 challenge leaderboard (Fang et al., 2022). As the leaderboard only reports the AUC metric, our comparison is limited to this metric.

As shown in Table 3, MedAgent-Pro outperforms these task-specific methods, despite the VLMs in MedAgent-Pro remaining zero-shot. In glaucoma diagnosis, the AUC metric has improved by 6.8%, while the bAcc and F1 scores have increased by 4.6% and 3.3%, respectively. This finding further demonstrates that integrating specialized tools with general VLMs can achieve performance comparable to domain-specific models, emphasizing the potential of the MedAgent-Pro framework.

| Setting | | | Glaucoma | | Heart Disease | |
|---|---|---|---|---|---|---|
| Planning | Action | Reflection | bAcc | F1 | bAcc | F1 |
| | | | 56.4 | 21.1 | 56.8 | 28.1 |
| ✓ | | | 75.9 | 36.5 | 63.3 | 45.9 |
| ✓ | ✓ | | 88.5 | 71.0 | 73.4 | 66.6 |
| ✓ | ✓ | ✓ | **90.4** | **76.4** | **77.8** | **72.3** |

Table 4: Ablation on key components, including Planning, Tool-based Action and Evidence-based Reflection.

| Method | bAcc | F1 |
|---|---|---|
| GPT-4o | 90.4 | 76.4 |
| VisionUnite | 92.9 | 79.1 |

Table 5: Ablation on qualitative indicators analysis.

### 4.3 ABLATION STUDY AND DETAILED ANALYSIS

**Effectiveness of the Proposed Key Components.** We conduct an ablation study on glaucoma and heart disease diagnosis to evaluate the effectiveness of three key modules: planning, tool-based action, and evidence-based reflection, while each module builds upon the previous one. As shown in Table 4, incorporating planning significantly improves overall performance, while integrating visual tools to support diagnostic action brings further gains, with F1 scores increasing by 34.5% and 20.7%, respectively. The addition of evidence-based reflection further enhances the consistency of plan execution and the reliability of actions, thereby reaching the best performance. These results validate the effectiveness of the components and demonstrate their complementary roles in jointly enabling evidence-based medical reasoning. Other detailed ablation of the toolset are in Appendix B.

**Impact of Qualitative and Quantitative Accuracy.** Both qualitative and quantitative analyses at each reasoning step may introduce errors. To assess their respective impact on final accuracy, we conduct ablation studies on glaucoma diagnosis. Since the accuracy of qualitative analysis is difficult to quantify, we use an ophthalmic-specific model VisionUnite (Li et al., 2024c) to conduct qualitative analysis instead of the original GPT-4o. As listed in Table 5, the VisionUnite only gains marginal improvement, indicating that general-purpose VLMs (e.g., GPT-4o), when guided by medical guidelines, are sufficient for qualitative analysis without requiring additional domain-specific tools. Furthermore, we simulate noisy segmentation masks following prior works (details in Appendix E), and observe that higher segmentation accuracy consistently yields better performance, as depicted in Fig. 5. These findings highlight the importance of tool-based actions in supporting quantitative analysis and the robustness gains achieved through evidence-based reflection.

Ablation on Tool Accessibility We further conduct an ablation study to demonstrate that the performance gains of MedAgent-Pro stem from its carefully-designed workflow that enhances agentic reasoning, rather than from merely integrating additional tools. In this experiment, the baseline models are given the same toolset directly, the results are shown below:

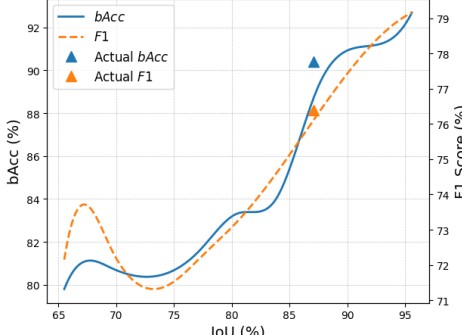

Figure 5: Ablation on quantitative indicator analysis that reveals how segmentation accuracy influences diagnostic outcomes.

| Baseline Model | Glaucoma | | Heart Disease | |
|---|---|---|---|---|
| | bAcc | F1 | bAcc | F1 |
| GPT-4o | 74.4 | 52.3 | 64.3 | 48.0 |
| Janus-Pro-7B | 62.6 | 39.4 | 56.8 | 36.4 |
| LLaVA-Med | 50.0 | 0.0 | 50.0 | 0.0 |
| Qwen2.5-7B-VL | 72.6 | 43.0 | 43.3 | 17.3 |
| InternVL2.5-8B | 60.6 | 27.7 | 51.4 | 13.3 |
| MedAgent-Pro (Ours) | **90.4** | **76.4** | **77.8** | **72.3** |

Table 6: Comparison of VLMs given with toolset access but without planning.

It can be seen that even when granted access to the toolset, these models fail to effectively utilize the tools for coherent reasoning. For example, despite having access to segmentation models for optic cup and disc and a coding module, they are unable to generate code to compute the cup-to-disc ratio.

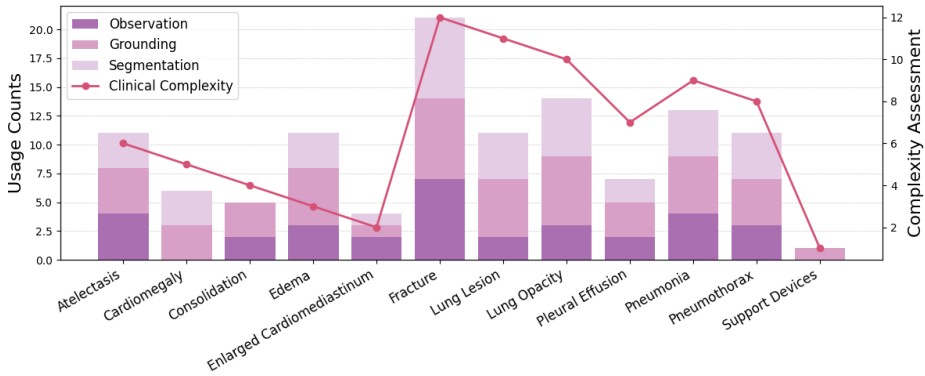

Figure 6: Diagnostic plan complexity vs. clinician assessment by subtasks on MIMIC.

## 5 CLINICAL EXPERT EVALUATIONS

**Alignment with Real-World Clinical Workflow.** To assess how well MedAgent-Pro aligns real clinical workflows, we quantified the number of steps it performs across 12 chest X-ray diagnostic tasks. Specifically, we compared the results with thoracic clinicians' rankings, where each task was rated from 1 to 12 based on perceived diagnostic difficulty and time demand. As shown in Fig. 6, most tasks show a clear positive correlation between total step count and physician-rated complexity. For instance, *Fracture* is the most complex and involves the highest number of steps, while *Support Devices* require the fewest step and is ranked least complex. Notably, conditions like *Pleural Effusion* and *Cardiomegaly* benefit from visual tool integration, significantly reducing workflow steps. In contrast, tasks such as *Fracture* and *Edema* remain step-intensive. This is because diagnoses relying on quantitative indicators (e.g., the cardiothoracic ratio) can be automated by visual tools, while those requiring qualitative assessment still depend on sequential reasoning. The findings demonstrate the effectiveness and practical compatibility of MedAgent-Pro's structured, evidence-based workflow with real-world clinical diagnostic process.

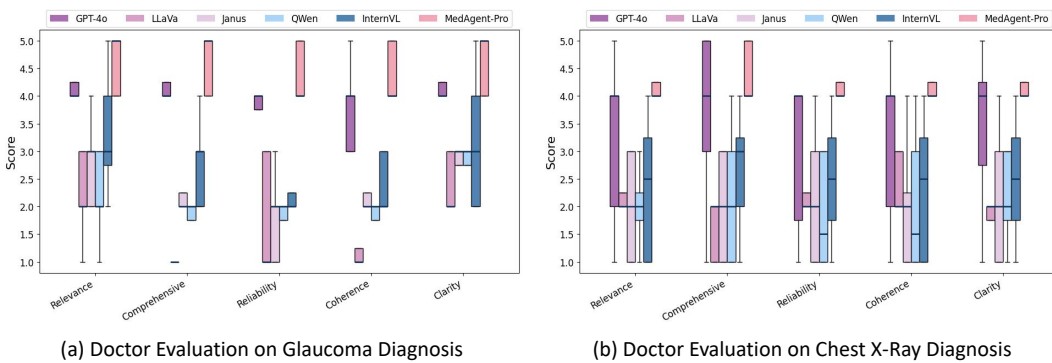

(a) Doctor Evaluation on Glaucoma Diagnosis      (b) Doctor Evaluation on Chest X-Ray Diagnosis

Figure 7: Evaluation on glaucoma and chest X-ray diagnosis by clinical experts.

**Assessment of Generated Diagnostic Content Across Other Methods.** To further assess diagnostic quality beyond accuracy, we conduct validation with clinical experts on glaucoma and chest X-ray diagnosis. The clinicians rate the diagnostic outputs from both VLMs and MedAgent-Pro across five dimensions—relevance, comprehensiveness, clinical reliability, reasoning coherence, and language clarity—using a 1 to 5 scale (details in Appendix E). MedAgent-Pro outperforms other VLMs across five dimensions and demonstrates strong stability across diverse cases, underscoring the alignment of our structured, evidence-based approach with modern medical diagnosis standards.

## 6 CONCLUSION

This paper introduces MedAgent-Pro, a reasoning agentic system designed to deliver accurate, multi-modal medical diagnoses. Addressing the limitations of current empirical diagnosis. This

represents a significant step toward the core principles of evidence-based medicine and advancing the practical application of AI in healthcare. However, the proposed framework depends on the availability of visual tools, which remain limited in certain medical domains. In addition, qualitative analysis still relies on VLMs, which are prone to VLMs' inherent hallucination. Addressing these limitations will further improve the reliability and clinical impact of computer-aided diagnosis.

## ACKNOWLEGEMENT

This work was supported by Tier 2 grant, Singapore (T2EP20224-0028), and Ministry of Education Tier 1 grant, NUS, Singapore (24-1250-P0001).

## THE STATEMENTS

### LLM USAGE STATEMENT

In this work, LLMs were used solely as writing assistants to polish the language of the manuscript. They were not involved in research ideation, experiment design, implementation, analysis, or any other scientific contribution.

### REPRODUCIBILITY STATEMENT

We have made extensive efforts to ensure the reproducibility of our work. The datasets used in our experiments are publicly available, and all preprocessing steps are described in detail in the main text and appendix. The implementation of MedAgent-Pro, including the workflow design, tool integration, and evaluation pipeline, has been uploaded to an anonymous GitHub repository at **https://anonymous.4open.science/r/MedAgent-Pro-CC0B**. Hyperparameter settings, model configurations, and evaluation metrics are fully documented in the appendix. Additional ablation studies and qualitative analyses are also provided in the appendix to further support reproducibility.

### ETHICS STATEMENT

This study involves human evaluation conducted by clinical experts. All participants were clearly informed of the purpose, scope, and potential risks of the study, and their participation was entirely voluntary with informed consent obtained beforehand. No sensitive patient data was collected or analyzed. The study was determined not to constitute human-subject research requiring IRB approval. We have adhered to the ICLR Code of Ethics throughout the study, ensuring fairness, transparency, and compliance with ethical and legal standards.

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

APPENDIX

**Table of content:**

## A  ADDITIONAL COMPARATIVE RESULT

Due to space limitations in the main paper, many of our results were not fully presented; these results are provided in the appendix below:

### A.1  THE FULL COMPARISON WITH DOMAIN-SPECIFIC METHODS ON MIMIC

In Table 3 of the main text, we only report the average balanced accuracy (bAcc) across the 12 tasks when comparing MedAgent-Pro with state-of-the-art (SOTA) chest X-ray diagnosis models such as CheXagent (Chen et al., 2024b) and Maira (Bannur et al., 2024). Therefore, we present the complete results for all 12 tasks below:

It can be observed from Table 7 that after fine-tuning on chest X-ray data, both models significantly outperform GPT-4o on most tasks, and exhibit exceptional performance on certain tasks. For example, CheXagent achieves a leading bAcc of 81.8% on *Lung Lesion* and 84.2% on *Pneumothorax*.

However, these models inevitably suffer from data imbalance, resulting in poor performance on certain datasets. For instance, CheXagent only achieves a bAcc of 43.3% on *Atelectasis*, and Maira reaches just 50.9% on *Pneumonia* diagnosis, which severely impacts their average performance.

Accordingly, in our proposed MedAgent-Pro, we construct a diagnosis plan for each disease using medical guidelines retrieved by the RAG agent, and utilize specialized visual tools to support professional analysis. This approach integrates domain knowledge directly into the workflow instead of fine-tuning the VLM, thereby avoiding the impact of data imbalance. As a result, our method achieves competitive performance across all 12 tasks, reaching the highest average bAcc of 72.0%.

| Method | Avg. | Atelectasis | Cardiomegaly | Consolidation | Edema | Enlarged Cardiomediastinum | Fracture |
|---|---|---|---|---|---|---|---|
| GPT-4o | 58.3 | 68.7 | 64.3 | 60.5 | 61.2 | 53.2 | 56.3 |
| Maira | 64.1 | 66.8 | 70.2 | 56.8 | 63.7 | 61.1 | 52.4 |
| CheXagent | 69.1 | 43.4 | 75.4 | 58.8 | **72.5** | 61.1 | **68.8** |
| MedAgent-Pro | **72.0** | **85.5** | 74.2 | **66.3** | 59.2 | **75.4** | 68.5 |

| Method | Avg. | Lung Leision | Lung Opacity | Pleural Effusion | Pneumonia | Pneumothorax | Supporting Devices |
|---|---|---|---|---|---|---|---|
| GPT-4o | 58.3 | 38.6 | 63.1 | 62.7 | 59.6 | 47.7 | 63.4 |
| Maira | 64.1 | 68.2 | **74.3** | 76.8 | 50.9 | 50.0 | 77.7 |
| CheXagent | 69.1 | **81.8** | 73.6 | **80.7** | **67.0** | **84.2** | 68.3 |
| MedAgent-Pro | **72.0** | 72.2 | 67.6 | 77.8 | 62.1 | 65.6 | **89.2** |

Table 7: Comparison with zero-shot foundation models on the MIMIC subset with task-specific models on bAcc metric (%). "Avg." is the average performance for the 12 sub-tasks.

## A.2 F1 SCORE RESULT ON THE MIMIC DATASET

Due to space constraints in the main text, we only reported bAcc results on the MIMIC dataset. Here, we provide the F1 scores across all 12 tasks for general VLMs, task-specific models, and the proposed MedAgent-Pro, along with their average.

| Method | Avg. | Atelectasis | Cardiomegaly | Consolidation | Edema | Enlarged Cardiomediastinum | Fracture |
|---|---|---|---|---|---|---|---|
| *VLM-based Methods* | | | | | | | |
| GPT-4o | 61.3 | 81.0 | 76.4 | 70.8 | **71.0** | 67.1 | 22.2 |
| Janus-Pro-7B | 28.5 | 37.6 | 39.3 | 14.8 | 24.0 | 15.4 | 0.0 |
| LLaVA-Med | 45.4 | 84.3 | 81.0 | **73.3** | 0.0 | 72.0 | 0.0 |
| BioMedClip | 50.8 | 82.9 | 66.2 | 68.2 | 0.0 | 36.4 | 0.0 |
| Qwen2.5-7B-VL | 40.4 | 56.4 | 65.2 | 60.5 | 36.9 | 0.0 | 0.0 |
| InternVL2.5-8b | 48.6 | 85.6 | 69.9 | 48.6 | 40.0 | 0.0 | 46.2 |
| *Agentic Systems* | | | | | | | |
| Maira | 43.6 | 50.3 | 61.5 | 24.0 | 49.1 | 36.3 | 16.3 |
| CheXagent | 70.3 | **92.5** | **81.3** | 48.5 | 69.4 | 36.4 | 54.5 |
| MedAgent-Pro | **73.0** | 83.1 | 80.3 | 58.8 | 66.7 | **82.4** | **71.4** |

| Method | Avg. | Lung Leision | Lung Opacity | Pleural Effusion | Pneumonia | Pneumothorax | Supporting Devices |
|---|---|---|---|---|---|---|---|
| *VLM-based Methods* | | | | | | | |
| GPT-4o | 61.3 | 37.5 | 80.2 | 81.4 | 67.2 | 0.0 | 80.7 |
| Janus-Pro-7B | 28.5 | 30.8 | 29.7 | 36.1 | 39.4 | 0.0 | 75.2 |
| LLaVA-Med | 45.4 | 0.0 | 81.2 | 83.1 | **70.0** | 45.9 | 76.0 |
| BioMedClip | 50.8 | 77.7 | 78.3 | **83.6** | 56.5 | 40.7 | 19.5 |
| Qwen2.5-7B-VL | 40.4 | 51.2 | 82.1 | 0.0 | 45.7 | 0.0 | 86.8 |
| InternVL2.5-8b | 48.6 | 30.8 | 77.4 | 57.3 | 35.8 | 14.9 | 77.0 |
| *Agentic Systems* | | | | | | | |
| Maira | 43.6 | 53.3 | 66.0 | 74.9 | 7.8 | 0.0 | 83.3 |
| CheXagent | 70.3 | **77.8** | **88.7** | 87.5 | 57.5 | **53.8** | 95.0 |
| MedAgent-Pro | **73.0** | 63.5 | 80.9 | 76.9 | 63.2 | 53.1 | **95.7** |

Table 8: Comparison with zero-shot foundation models and domain-specific models on the MIMIC subset on F1 score (%). "Avg." is the average performance for the 12 sub-tasks.

As shown in Table 8, our proposed MedAgent-Pro achieves state-of-the-art performance on the MIMIC dataset, attaining the highest overall average F1 score of 73.0% across all 12 diagnostic tasks. It outperforms both general-purpose VLMs (e.g., GPT-4o at 61.3%) and task-specific models (e.g., CheXagent at 72.1%), and demonstrates exceptional results on key clinical tasks such as Enlarged Cardiomediastinum and Fracture. Unlike other models that suffer from performance degradation due to data imbalance or limited task specialization, MedAgent-Pro maintains consis-

tently strong results across all tasks, with its lowest F1 score still reaching 53.1% on Pneumothorax, highlighting its robustness and generalizability.

While the F1 score is a widely used metric for evaluating classification performance, it also has inherent limitations, particularly in imbalanced datasets. The F1 score emphasizes performance on the positive class and may overlook the model's ability to correctly identify negative cases. As a result, a model can achieve a deceptively high F1 score by predicting all instances as positive. For instance, on the Pleural Effusion task, LLaVA-Med achieves the second-highest F1 score (83.1%) among all models. However, its balanced accuracy (bAcc) is only 50.0%, indicating that the model predicts all samples as positive without discriminating between classes. Therefore, to fairly and comprehensively assess a model's performance, the F1 scores presented in Table 8 should be interpreted alongside the bAcc metrics reported in Table 2 of the main text.

## A.3 RESULTS ON ADDITIONAL METRICS

For the binary diagnosis tasks like glaucoma and heart disease diagnosis, we also report recall and precision values here to offer a more comprehensive view of the model's performance. It can be seen that our model not only performs well on bAcc and F1 metric, but also have significant performance gain on the recall and precision metric, which shows our method presents high generalization ability across different tasks.

| Method | Glaucoma | | | | Heart Disease | | | |
|---|---|---|---|---|---|---|---|---|
| | bAcc | F1 | Recall | Precision | bAcc | F1 | Recall | Precision |
| *VLM-based Methods* | | | | | | | | |
| GPT-4o | 56.4 | 21.1 | 30.0 | 16.2 | 56.8 | 28.1 | 17.3 | 75.0 |
| Janus-Pro-7B | 53.4 | 13.3 | 7.5 | 60.0 | 52.3 | 10.7 | 5.7 | 75.0 |
| LLaVA-Med | 50.0 | 0.0 | 0.0 | 0.0 | 50.0 | 0.0 | 0.0 | 0.0 |
| BioMedClip | 58.1 | 21.3 | 75.0 | 12.4 | 47.0 | 37.8 | 40.3 | 35.6 |
| Qwen2.5-7B-VL | 54.3 | 16.3 | 10.0 | 44.4 | 50.0 | 0.0 | 0.0 | 0.0 |
| InternVL2.5-8B | 51.8 | 13.8 | 15.0 | 12.8 | 49.7 | 3.6 | 1.9 | 33.3 |
| *Agentic Systems* | | | | | | | | |
| MedAgents (ACL'24) | 52.1 | 8.9 | 5.0 | 40.0 | 51.1 | 15.9 | 9.6 | 45.5 |
| MMedAgent (EMNLP'24) | 52.4 | 16.3 | 25.0 | 12.0 | 55.0 | 26.7 | 17.3 | 60.0 |
| MDAgent (NeurIPS'24) | 56.8 | 22.2 | 22.5 | 22.0 | 57.2 | 30.3 | 19.2 | 71.4 |
| MedAgent-Pro (Ours) | **90.4** | **76.4** | **85.0** | **69.4** | **77.8** | **72.3** | **65.4** | **81.0** |

Table 9: Comparison with general VLMs and medical agentic systems on REFUGE2 (Glaucoma) and MITEA (Heart Disease) datasets with additional metrics (%).

## A.4 RESULTS OF SUB-TASKS IN THE HEART DISEASE DIAGNOSIS

In the main text, we simplify the heart disease task in the Mitea dataset into a binary classification task. Here we present the full result on all the sub-task in the Mitea dataset, which is amyloidosis, aortic regurgitation, cardiomyopathy dilated, cardiomyopathy hypertrophic, hypertrophy and transplant. The results are shown below:

| Method | Avg. | Amyloidosis | Aortic Regurgitation | Cardiomyopathy Dilated | Cardiomyopathy Hypertrophic | Hypertrophy | Transplant |
|---|---|---|---|---|---|---|---|
| Cases | / | 12 | 10 | 6 | 8 | 14 | 2 |
| GPT-4o | 56.8 | 65.6 | 50.6 | 58.6 | 60.5 | 52.4 | 53.0 |
| Janus-Pro-7B | 49.3 | 57.6 | 35.8 | 49.1 | 53.9 | 64.6 | 34.5 |
| LLaVA-Med | 51.6 | 50.0 | 54.1 | 49.6 | 55.9 | 50.0 | 50.0 |
| Qwen2.5-7B-VL | 49.4 | 48.8 | 50.0 | 49.6 | 48.9 | 50.0 | 48.9 |
| InternVL2.5-8B | 52.5 | 54.9 | 51.8 | 62.6 | 48.1 | 48.5 | 48.9 |
| MedAgent-Pro | **82.8** | **93.0** | **80.2** | **87.4** | **85.1** | **80.7** | **70.5** |

Table 10: The bAcc value of the sub-tasks in heart disease diagnosis.

As shown, MedAgent-Pro achieves even better performance when diagnosing more specific heart-disease subtypes, substantially outperforming VLMs. This is because a clearer disease query allows MedAgent-Pro to focus on more direct clinical indicators (e.g., ventricular wall thickening for amyloidosis), leading to more accurate decisions.

## B ADDITIONAL ABLATION STUDIES

### B.1 ABLATION ON THE RAG AGENT

LangChain was employed in our RAG agent implementation mainly for convenience, as it offers a modular and widely adopted interface. Importantly, the retrieval module is fully interchangeable, and our framework is not tied to any specific implementation. To verify this, we replaced LangChain's retriever with a FAISS-based alternative, which is fully open-source and independent of LLMs. The results are shown below:

| RAG Method | Glaucoma | | Heart Disease | |
|---|---|---|---|---|
| | bAcc | F1 | bAcc | F1 |
| FAISS | 91.5 | 76.9 | 77.2 | 71.6 |
| LangChain (Ours) | 90.4 | 76.4 | 77.8 | 72.3 |

Table 11: The impact of different implementations of the RAG agent.

As shown in the table, switching to an open-source retrieval method leads to minimal performance degradation and even slight improvements in glaucoma diagnosis, demonstrating that our framework is robust to the underlying RAG implementation.

As for the exploration of hypersparameters in the RAG module, we conducted an ablation study on the number of retrieved chunks $k$ used for generating the diagnostic plan:

| $k$ for Top-$k$ | Glaucoma | | Heart Disease | |
|---|---|---|---|---|
| | bAcc | F1 | bAcc | F1 |
| 1 | 84.7 | 80.0 | 70.7 | 61.1 |
| 2 | 87.6 | 78.5 | 71.7 | 62.8 |
| 5 | 90.4 | 76.4 | 77.8 | 72.3 |
| 10 | 90.4 | 76.4 | 77.2 | 71.5 |

Table 12: The impact of the hyperparameter in the RAG agent.

As shown in the table, our framework exhibits strong robustness to variations in the top-$k$ retrieval parameter. While a larger $k$ offers marginal improvements up to a point, the overall performance remains stable across a wide range.

We have further replaced MedlinePlus with **ACC/AHA guideline documents** for the Heart Disease task, which is more professional guidelines from medical societies for heart disease diagnose. The results are shown in Table 13.

| RAG source | Heart Disease bAcc | Heart Disease F1 |
|---|---|---|
| MedlinePlus (Default) | 77.8 | 72.3 |
| ACC/AHA Guidelines | 77.2 | 71.5 |

Table 13: The impact of different RAG knowledge base.

It can be seen that using ACC/AHA guidelines as the knowledge base for RAG can slightly improve the overall performance, further demonstrating that MedAgent-Pro is a flexible framework, and can improve performance with more professional knowledge base or tools.

We attribute this stability to two factors: (1) The structured workflow and tool-guided reasoning play a dominant role in diagnosis quality; (2) Given that current VLMs have been trained on knowledge bases with medical knowledge like Wikipedia, they are often able to generate reasonable diagnostic plans with intrinsic knowledge even when the retrieved evidence slightly varies.

Noted that incorporating hospital-specific guideline or private RAG sources is feasible within our framework, as such customization further enhances the clinical relevance of MedAgent-Pro.

## B.2    ABLATION ON DIFFERENT TOOLSET

We have further evaluated the performance on toolset variety, including visual analysis models, coding modules, and qualitative reasoning models (qualitative models **were not used** in the main text due to their relatively limited improvement and strong domain specificity).

| Setting | | | Glaucoma | |
|---|---|---|---|---|
| Visual Tools | Coding Model | Qualitative Models | bAcc | F1 |
| | | | 56.4 | 21.1 |
| ✓ | | | 87.0 | 55.0 |
| | | ✓ | 78.9 | 38.5 |
| ✓ | ✓ | | 90.4 | 76.4 |
| ✓ | ✓ | ✓ | 92.9 | 79.1 |

Table 14: Ablation on key components with different combinations.

The results still support the robustness of our workflow design. Since the qualitative analysis tools (e.g., domain-specific VLMs) are hard to obtain for all diagnosis tasks and can be done by VLMs themselves, we do not include them in the workflow.

## B.3    ABLATION ON TOOL ACCESSIBILITY

Meanwhile, we further demonstrate MedAgent-Pro's effectiveness even without access to any external tools, using only retrieved knowledge and structured reasoning. In cases in the NEJM dataset where no external tools are accessible, the results are shown below:

| Method | Acc |
|---|---|
| ***VLM-based Methods*** | |
| GPT-4o | 74.7 |
| Janus-Pro-7B | 29.6 |
| LLaVA-Med | 25.2 |
| BioMedClip | 27.3 |
| Qwen2.5-7B-VL | 42.9 |
| InternVL2.5-8B | 44.0 |
| ***Agentic Systems*** | |
| MedAgents | 69.3 |
| MMedAgent | 75.3 |
| MDAgent | 77.5 |
| MedAgent-Pro (Ours) | **84.0** |

Table 15: Comparison of methods on the NEJM cases where no suitable visual tools are available.

As shown in the table, our method still achieves the best performance, highlighting that the reasoning structure of MedAgent-Pro is capable of leveraging tool outputs when available, and maintaining strong performance even when tool access is limited due to the system's planning capability and integration of retrieved knowledge.

### B.4 ANALYSIS OF DECISION-MAKING STRATEGIES.

We further explore alternative decision-making strategies in the risk-based decision stage. In addition to our proposed *structured fusion*, which assigns risk-based weights to clinical indicators, we evaluate *flat fusion*, where all raw indicators are directly fed into the VLM As illustrated in Fig. 8, structured fusion consistently outperforms flat fusion across varying indicator counts, leading to more balanced and comprehensive decisions, whereas VLMs often focus on partial cues.

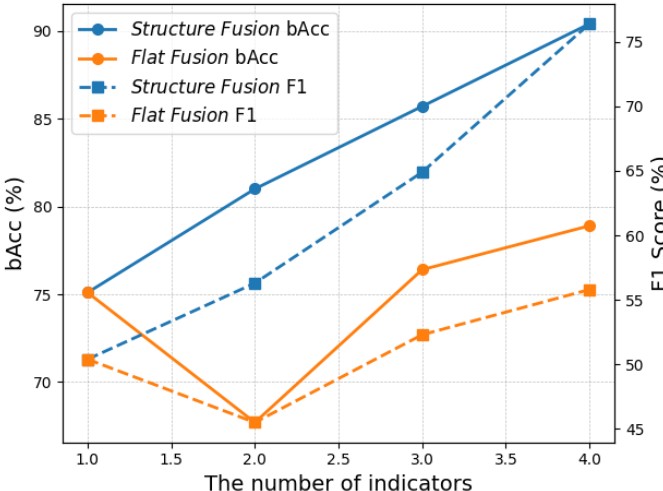

Figure 8: Comparison of two kinds of decision-making ways on Glaucoma diagnosis.

## C OTHER DETAILED ANALYSIS

### C.1 STUDY ON COMPUTATION COST

Regarding computational cost, our workflow is actually efficient: for each disease, the planning stage is executed only once across all patients, generating reusable diagnostic code and workflows. Below, we report the average inference time per case (evaluated on an A6000 GPU):

As shown, MedAgent-Pro achieves competitive speed compared to existing medical agentic systems. The overhead from planning and code execution is minimal (typically under 1 second), and the majority of runtime is spent on the backbone VLM, similar to other methods.

### C.2 DISCUSSION ON RELATION TO MEDICAL WORLD MODEL

Recently, the Medical World Model (Yang et al., 2025) has attracted attention for its ability to provide visual evidence through generative simulation for treatment planning. Here, we discuss its relation to MedAgent-Pro and clarify the differences between the two approaches.

Regarding visual evidence, MedAgent-Pro directly integrates visual envidence through specialized quantitative tools. For example, the glaucoma workflow uses segmentation models to obtain optic disc and cup region, and in the chest X-ray diagnosis, grounding tools can show the location of abnormalities like effusions.

Meanwhile, MeWM generates visual evidence through a generation model that simulates disease progression under different conditions to support treatment decisions. Although the tasks and mechanisms differ, both approaches incorporate visual evidence aligned with their respective objectives.

| Method | time (s) |
|---|---|
| ***VLM-based Methods*** | |
| GPT-4o | 21.2 |
| Janus-Pro-7B | 14.1 |
| LLaVA-Med | 1.5 |
| Qwen2.5-7B-VL | 5.1 |
| InternVL2.5-8B | 10.7 |
| ***Agentic Systems*** | |
| MedAgents | 143 |
| MMedAgent | 32 |
| MDAgent | 272 |
| MedAgent-Pro (Ours) | 44.6 |

Table 16: Study on computational cost on Glaucoma diagnosis.

## D    BACKGROUND OF EVIDENCE-BASED MEDICINE

Our proposed MedAgent-Pro is inspired by the principle of evidence-based medicine in modern healthcare. Here, we provide a detailed introduction to this concept to help readers better understand the design rationale behind our agentic workflow.

**Background of Evidence-based Medicine.** Evidence-Based Medicine (EBM) is defined as "the conscientious, explicit, and judicious use of current best evidence in making decisions about the care of individual patients. It means integrating individual clinical expertise with the best available external clinical evidence from systematic research" (Guyatt et al., 1992a). The goal of EBM is to integrate clinical expertise, patient values, and the best available scientific evidence to guide decisions related to clinical management. The term was originally used to describe an approach to teaching medical practice and improving individual physicians' decision-making for individual patients (Guyatt et al., 1992b). Adoption of evidence-based medicine is necessary in a human rights-based approach to public health and a precondition for accessing the right to health (Beracochea et al., 2010).

The term "evidence-based" was first used by David M. Eddy in 1987 in workshops and a manual commissioned by the Council of Medical Specialty Societies to teach formal methods for designing clinical practice guidelines (Eddy, 1990d). In 1996, David Sackett and colleagues clarified the definition of EBM as "the conscientious, explicit, and judicious use of current best evidence in making decisions about the care of individual patients." This involves integrating individual clinical expertise with the best available external evidence from systematic research (Grobbee & Hoes, 2014). EBM aims to make individual clinical decisions more structured and objective by grounding them in research evidence. At the individual level, EBM is further defined as the process of finding, appraising, and applying contemporaneous research findings as the basis for medical decisions (Katz, 2001).

**Link with the proposed MedAgent-Pro Workflow.** In the MedAgent-Pro workflow, we employ an RAG agent at the disease level to retrieve relevant medical guidelines, ensuring that clinical expertise is explicitly incorporated into the workflow. Meanwhile, we combine the medical guideline with the available toolset to construct disease-specific plans, which integrate evidence into the reasoning process meaningfully, rather than merely aggregating facts. This approach guides a step-by-step reasoning in a clear and structured manner, aligning with the EBM concept.

At the patient level, the evidence-based reasoning process follows disease-specific plans, conducting both qualitative and quantitative analyses step by step to enable comprehensive assessment and decision-making. Furthermore, each reasoning step is followed by a verification process to ensure that only clinically valid medical data is incorporated into the final decision. This design promotes the practical implementation of the EBM paradigm within the AI diagnostic workflow.

# E    IMPLEMENTATION DETAILS

## E.1    EXPERIMENTAL SETUP

In this section, we provide additional experimental details that were omitted from the main text due to space constraints. **Datasets.** For glaucoma diagnosis, we use the REFUGE2 dataset (Fang et al., 2022), which is a 2D retinal fundus image dataset containing 1200 RGB images at a resolution of $2124 \times 2056$ annotated by experts. Each image includes segmentation masks for the optic disc and optic cup, along with a classification label indicating whether the patient has glaucoma.

For heart disease diagnosis, we utilize the MITEA dataset (Zhao et al., 2023), a 3D echocardiography dataset comprising 536 images from 143 human subjects. Each image contains segmentation masks for the left ventricular myocardium and cavity, along with a classification label identifying the patient's heart condition across seven categories: healthy, aortic regurgitation, dilated cardiomyopathy, amyloidosis, hypertrophic cardiomyopathy, hypertrophy, and transplant. Due to the limited number of samples for each specific heart condition (some may only have 2 or 3 cases), we simplify the task to a binary classification: determining whether the patient has heart disease.

For chest X-ray diagnosis, we use the MIMIC dataset (Johnson et al., 2023), which includes 2D chest X-rays labeled with 12 potential abnormalities, such as Atelectasis, Cardiomegaly, Edema, Enlarged Cardiomediastinum, Fracture, Lung Leision, Lung Opacity, Pleural Effusion, Pneumonia, Pneumothorax and Supporting Devices. We sample 442 chest X-ray cases from 100 patients due to the large scale of the original dataset.

Meanwhile, we further employ New England Journal of Medicine (NEJM) database (of Medicine), where we compile 992 real-world diagnostic cases, encompassing over 10 anatomical regions, 10 imaging modalities, and 50 diseases. We present several examples here to provide a more intuitive understanding as below:

**Evaluation Metrics.** We make a detailed description of the metrics we used in the main paper: The bAcc is defined as $\frac{1}{2}(\frac{TP}{TP+FN} + \frac{TN}{TN+FP})$, the F1 score is calculated as $2\frac{Precision \times Recall}{Precision+Recall}$.

**Dealing with MCQ questions** Clinicians in practice typically narrow down to a small set of suspected conditions before further differentiating among them Falik (2006), MedAgent-Pro generally follows this realistic workflow. In datasets like NEJM, questions are given in the format of multi-choice questions (MCQ). MedAgent-Pro constructs a diagnostic plan for each candidate option, performs analysis for each disease separately, and then lets the VLM integrate the resulting evidence and indicators to choose the most plausible answer.

## E.2    GENERATION OF NOISY LABELS

In Section 4.3 of the main text, we discuss how inaccurate segmentation results can affect the final diagnostic accuracy. Due to space constraints, we provide a detailed description of the noise label generation process here.

To simulate noisy annotations, we implement three types of label perturbations—dilation, erosion, and affine transformation—to simulate typical deviations found in manual annotations following (Xue et al., 2020; Zhang et al., 2020a;b; Zhu et al., 2019). For each training sample, one noise type is randomly applied with a noise intensity controlled by the parameter $\beta \in (0, 1)$, which represents the proportion of desired label distortion relative to the original foreground area.

**Dilation-based Noise.** Given a binary label $L \in \{0, 1\}^{W \times H}$, we define the foreground area as $|L| = \sum L$. The target noisy label $\tilde{L}$ satisfies:

$$|\tilde{L}| \approx |L| + \beta \cdot |L|$$

Iterative binary dilation is applied until the foreground area reaches $(1 + \beta) \cdot |L|$. The final mask is chosen to be the one closest to this target ratio while avoiding overflow beyond the image boundaries.

**Erosion-based Noise.** Erosion is used to simulate under-segmentation. The goal is to reduce the foreground such that:

$$|\tilde{L}| \approx (1 - \beta) \cdot |L|$$

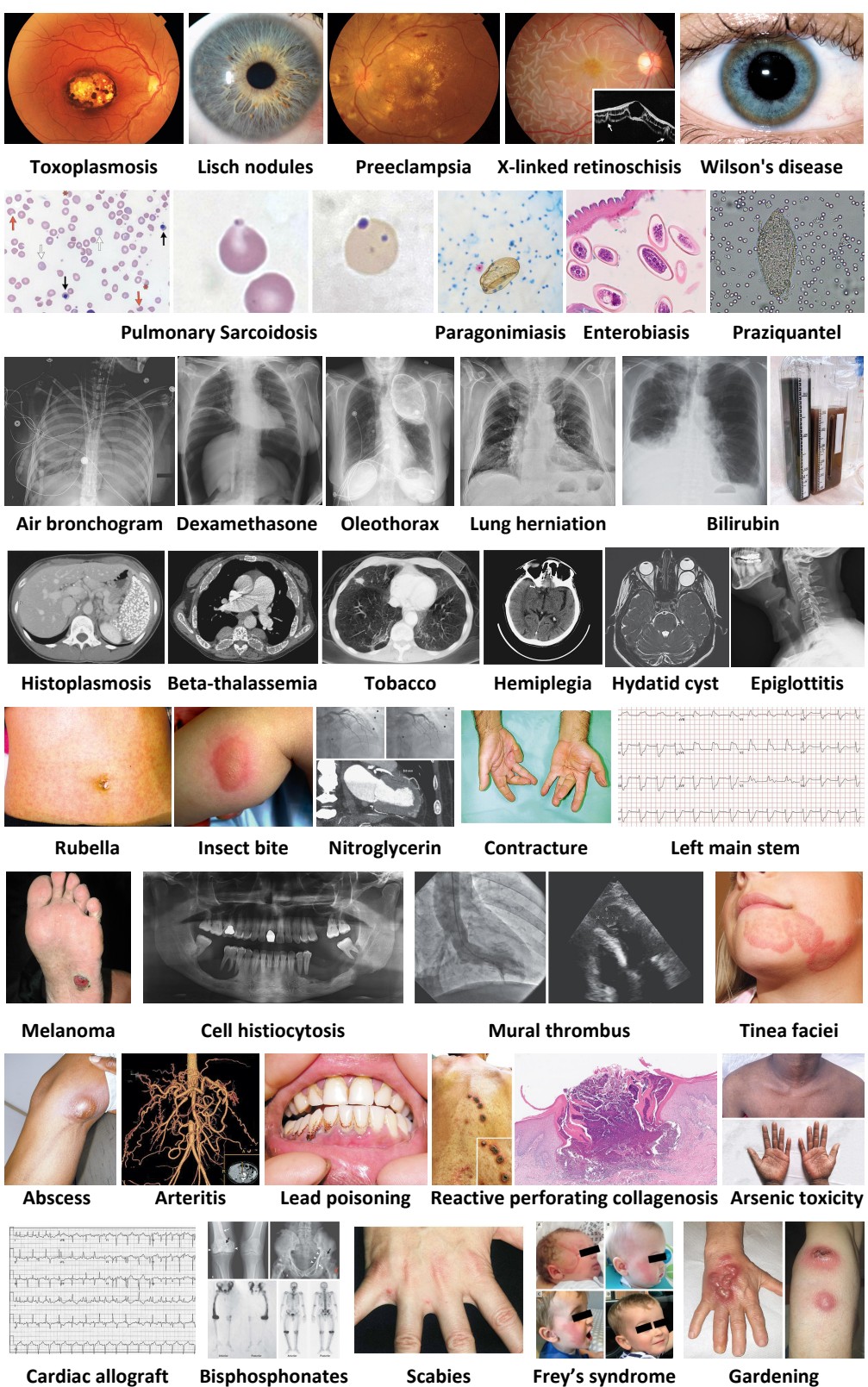

Figure 9: Example diagnostic cases from the NEJM dataset.

Binary erosion is applied iteratively until the foreground shrinks to the target level, again selecting the mask closest to the target size.

**Affine Transformation-based Noise.** This simulates geometric misalignments (e.g., due to inconsistent viewing windows). The mask is rotated by a random angle $\theta \in [-\theta_{\max}, \theta_{\max}]$ and then translated by an adaptive step $s$ along a randomly chosen direction. The amount of introduced noise is quantified as:

$$\text{NoiseRate} = \frac{|\tilde{L} \oplus L|}{|L|}$$

where $\oplus$ denotes the symmetric difference between the original and transformed masks. A binary search is used to find the smallest step $s$ such that $\text{NoiseRate} \approx \beta$, allowing fine-grained control over the perturbation strength.

The parameter $\beta$ directly determines the *target deviation ratio* between the noisy and original mask, effectively simulating varying degrees of annotation error. It ensures that the noise is proportional to the original label size, making the corruption **scale-aware** and clinically realistic. A higher $\beta$ simulates more aggressive annotation noise. Below are examples of some noisy labels:

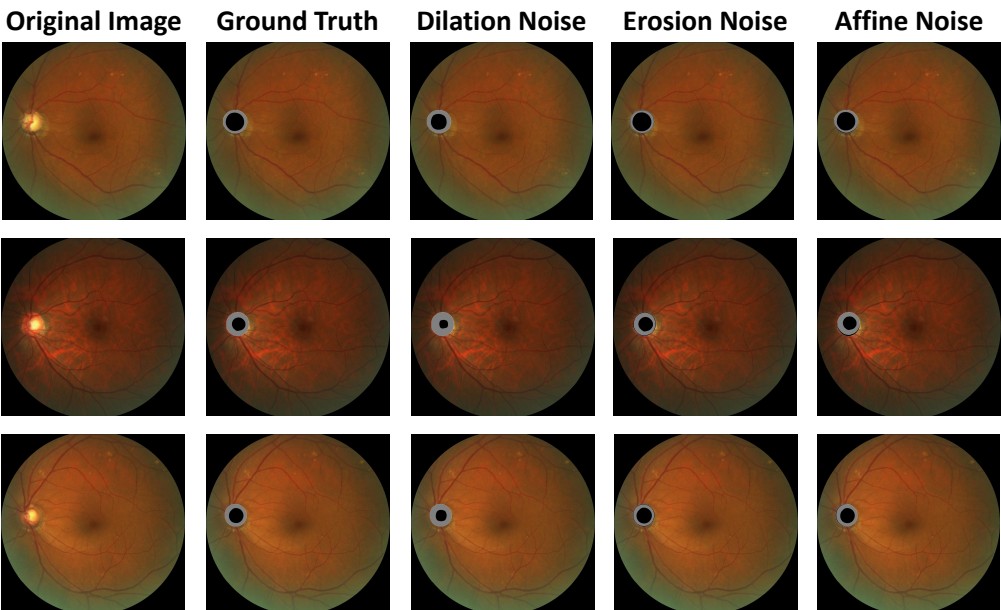

Figure 10: Example of noisy labels with different noise types

### E.3    CRITERIA FOR HUMAN EVALUATION

The clinicians rate the diagnostic outputs from both VLMs and MedAgent-Pro across five dimensions—relevance, comprehensiveness, clinical reliability, reasoning coherence, and language clarity—using a 1 to 5 scale following previous standards.

**Relevance (Content Relevance).** Relevance evaluates how accurately the model's response reflects the content and clinical significance of the input image and question (Can et al., 2025; Tam et al., 2024). A relevant response should focus on visible anatomical or clinical features and stay aligned with the diagnostic task, avoiding unrelated or off-topic information. The assessment considers whether the response directly references key image findings, maintains contextual focus, and excludes irrelevant details. Responses are rated on a scale from 1 to 5, with 5 indicating fully relevant and image-grounded observations, and 1 indicating a largely irrelevant or off-topic reply.

**Comprehensiveness (Coverage of Factors).** Comprehensiveness assesses whether the model's response takes into account all pertinent clinical features necessary for a well-rounded and accurate diagnosis (Hershenhouse et al., 2024; Singhal et al., 2023). It evaluates the inclusion of a broad range of relevant factors, such as differential diagnoses, comorbidities, and key clinical indicators.

A comprehensive response should reflect the depth and balance expected from a human expert, without overemphasizing a single feature. Scores range from 1 to 5, with 5 indicating a thorough and balanced discussion of all clinically relevant aspects, and 1 reflecting a shallow response lacking diagnostic depth.

**Clinical Reliability (Expertise).** Clinical reliability evaluates the extent to which the response aligns with medical standards, evidence-based medicine practices, and expert clinical understanding (Fast et al., 2024). A correct answer should be accurate, use appropriate terminology, and reflect reasoning that would be trusted in a clinical context. The response is assessed based on its consistency with established guidelines and expert consensus. Scores range from 1 to 5, where 5 denotes expert-level accuracy and full alignment with clinical standards, and 1 indicates clinically incorrect or misleading content.

**Reasoning Coherence (Logical Consistency).** Reasoning Coherence evaluates whether the model's diagnostic output follows a coherent and logical clinical reasoning process (Hershenhouse et al., 2024; Singhal et al., 2023). A correct conclusion alone is insufficient—what matters is that the explanation demonstrates sound medical logic, rather than guesswork or unsupported assumptions. This criterion considers whether diagnostic steps are appropriate, clinical inferences are well-justified, and the overall reasoning pathway is traceable and explainable. Scores range from 1 to 5, with 5 indicating fully logical, medically justified reasoning, and 1 reflecting an illogical or fundamentally flawed diagnostic process.

**Language Clarity.** Language clarity gauges how effectively the model communicates its reasoning and conclusions (Can et al., 2025; Tam et al., 2024; Hershenhouse et al., 2024). The explanation should use professional, clinically appropriate language that is also clear and accessible to the intended audience. This metric evaluates whether the language is precise and unambiguous, medical terminology is correctly used, grammar and structure are sound, and whether a professional reader can easily follow the rationale without confusion. Scores range from 1 to 5, where 5 denotes crystal-clear and professional expression, and 1 reflects poor language quality that hinders understanding.

**Evaluation Details** Two licensed clinicians participated in the study: one ophthalmologist evaluated the glaucoma cases, and one thoracic surgeon evaluated the chest X-ray cases. Each rater assessed only cases within their clinical specialty, ensuring domain-appropriate judgment.

To mitigate anchoring or exposure effects, we used a randomized and blinded evaluation protocol. For each task, all model outputs (MedAgent-Pro and comparision methods) were anonymized, and both the case order and the ordering of outputs for each case were randomly shuffled for the clinician. This prevents bias arising from fixed sequencing or repeated exposure.

### E.4 Details of Evidence-based Reflection

For completeness, we provide a more detailed description of the Evidence-Based Reflection mechanism used in patient-level reasoning. The state-assessment function $\phi$ is a zero-shot evaluator implemented by the VLM without any training. Specifically, as discussed in Section 3.3, each executed step $P_i'$ produces a result $r_i$ from the input object $o_i$, which can be (i) a visual model output (e.g., segmentation mask or bounding boxes), (ii) a numerical result (e.g., the cup-to-disc ratio), or (iii) a qualitative analysis. The function $\phi$ evaluates the reliability of the result from the following aspects:

**(1) Input-quality assessment.** The reflection prompt explicitly asks the VLM to judge the quality of the input image and the feasibility of performing the specified action (e.g., "segment optic disc" or "assess pleural effusion"). The VLM evaluates whether the image is blurry, occluded, cropped, low-resolution, or missing the relevant anatomical region.

**(2) Visual model output assessment.** If $r_i$ is the output of a visual tool, the VLM is prompted to assess whether the predicted region is anatomically plausible. The VLM examines (i) whether the mask or bounding box has a reasonable size (e.g., masks or boxes that cover nearly the entire image are clearly incorrect), (ii) whether the shape is reasonable (e.g., the mask contains large holes or appears fragmented), and (iii) whether the prediction is consistent with the clinical guideline (e.g., "the cup region should lie inside the disc region").

**(3) Numerical-result assessment.** When $r_i$ is a numerical value (e.g., vCDR ratio or LVEF estimate), $\phi$ checks whether the value lies within a physiologically valid range based on the guideline $G$ (e.g., the vCDR value should be less than 1).

**(4) Qualitative textual assessment.** When $r_i$ is a qualitative statement produced by the VLM or a grounding tool (e.g., "mild opacity in the right lower lobe"), the reflection prompt requires that such judgments be made only when clear visual or textual clues are present. If the VLM expresses uncertainty (e.g., "the abnormality is not clearly visible"), the result is considered unreliable.

We do not train this module for two reasons: (1) annotated data for determining the reliability of each result is difficult to obtain, and (2) training such a component would undermine the zero-shot fairness of our comparisons. Our goal is to provide a flexible workflow that can adapt to a wide range of clinical scenarios. Despite being zero-shot, the ablation studies in the main paper demonstrate that the carefully designed evidence-based reflection mechanism is effective.

**Upper Bound in Evidence-based Reasoning** During runtime in the patient-level evidence-based reasoning, we set a strict upper limit on the reasoning rounds 10. When this limit is reached, the corresponding branch is terminated and excluded from the final decision-making to ensure stability.

### E.5 DETAILS OF THE TOOLSET

We have added the detailed description of the tools used in MedAgent-Pro as below. As shown in the table, only the glaucoma and heart-disease datasets provide ground-truth masks, allowing us to finetuning model or quantitatively evaluate the performance of the tools. All other tools are used in a zero-shot manner, please refer to their original papers for their general performance, as these tasks lack annotation for quantitative assessment in our tasks.

| Usage | Task | Tool / Model | If Trained |
|---|---|---|---|
| Segmentation | Glaucoma | Medical SAM-Adapter | Yes |
| | Heart Disease | Medical SAM-Adapter | Yes |
| | MIMIC-CXR | MedSAM | No |
| | NEJM Chest X-ray | MedSAM | No |
| | NEJM CT & MRI | MedSAM | No |
| | NEJM Cell Imaging | Cellpose | No |
| Grounding | MIMIC-CXR | Maira-2 | No |
| Coding / Utility Tools | All tasks | GitHub Copilot (code execution / parsing) | No |

Table 17: The details of the toolset used in MedAgent-Pro.

## F ADDITIONAL QUALITATIVE ANALYSIS

Due to space limitations in the main text, we present additional comparative experiments here, including results for both general VLMs and MedAgent-Pro on glaucoma diagnosis and chest X-ray diagnosis tasks. It is worth noting that we omit overly lengthy reasoning steps and redundant or meaningless repetitions from the presented content.

The comparison shows that, unlike general VLMs, the proposed MedAgent-Pro not only produces accurate conclusions but also demonstrates a more comprehensive reasoning process, according to its integration of an RAG agent to generate diagnostic plans based on medical guidelines. Meanwhile, its quantitative analysis grounded in evidence-based reasoning further enhances the reliability and trustworthiness of the final diagnosis.

| | | |
|---|---|---|
| **Original Image** |  |  |
| **GPT-4o** | The fundus image shows possible signs of glaucomatous optic nerve damage, most notably increased optic cup size and possible rim thinning. However, this image alone is not sufficient to definitively diagnose glaucoma, and further clinical evaluation and diagnostic testing are necessary.

Recommendation: Refer to an ophthalmologist for a comprehensive glaucoma workup, including IOP measurement, visual field testing, and OCT imaging. | This fundus image shows the posterior segment of the eye, including the optic disc, the macula, and retinal blood vessels. To assess for glaucoma, we need to consider several features, especially related to the optic nerve head (optic disc appearance):......

This patient may have signs suggestive of glaucoma based on:  A. large cup-to-disc ratio. Thinning of the neuroretinal rim, particularly inferiorly. |
| **Janus-Pro-7B** | The image provided is a fundus photograph, which is a detailed image of the retina at the back of the eye. Here are the key observations:

1. Optic Disc: The bright circular area on the left side of the image is the optic disc, where the optic nerve exits the eye. 2. Blood Vessels: Radiating from the optic disc are several retinal blood vessels. These vessels appear as fine, red lines. 3. Retinal Layers: The image shows the layered structure of the retina, with the photoreceptor layer at the bottom and the ganglion cell layer at the top.

General Health: The overall appearance of the retina seems normal, with no obvious signs of disease or abnormalities. | The image provided is a fundus photograph, which is a detailed image of the retina at the back of the eye. Here are the key observations:

1. Normal Retina: The majority of the image shows a normal retina with a reddish-brown background and visible blood vessels. 2. Lesion or Abnormality: There is a noticeable bright yellow area on the left side of the image, which appears to be an abnormal spot or lesion.3. Blood Vessels: The retinal blood vessels are visible and appear to be normal in size and pattern.

General Health: The overall health of the retina appears to be normal, with no signs of retinal detachment or other significant abnormalities. |
| **LLaVA-Med** | According to the fundus image, the patient does not have glaucoma. The image shows a normal optic nerve, which is an important indicator of the absence of glaucoma. | According to the fundus image, the patient does not have glaucoma. The image shows a normal optic nerve, which is an important indicator of the absence of glaucoma. |
| **Qwen2.5-7B-VL** | The provided fundus image shows a retinal view of an eye, but it does not provide enough detail to definitively diagnose glaucoma. Here are some key points to consider:

1. Optic Nerve Head (ONH): The optic nerve head appears to be within normal limits in terms of color and cup-to-disc ratio. However, this is just one aspect of glaucoma diagnosis.

2. Retinal Blood Vessels:The retinal blood vessels appear to be normal in caliber and distribution. There are no signs of significant narrowing or dilation that would suggest vascular changes associated with glaucoma. | The provided fundus image shows a retinal view of an eye, but it does not provide enough detail to definitively diagnose glaucoma. Here's why:

1. Optic Nerve Head (ONH) Appearance: The optic nerve head in this image appears relatively normal without any visible signs of cupping or pallor that might suggest glaucomatous damage. However, a definitive assessment of the optic nerve requires a stereoscopic view and a comparison with the fellow eye.

2. Retinal Blood Vessels: The retinal blood vessels appear to be within normal limits, neither excessively dilated nor constricted |
| **InternVL-2.5-8B** | To determine if the patient has glaucoma based on the fundus image, we need to examine the optic disc, the optic cup, and the surrounding retinal nerve fiber layer. Glaucoma is typically associated with optic disc changes, such as increased cup-to-disc ratio, rim thinning, and/or the presence of a rim excavation.

Based on the image provided, there are no obvious signs of glaucoma. The optic disc and optic cup appear normal, the RNFL is thick and continuous, and there is no visible peripapillary atrophy or blood vessel changes. However, a definitive diagnosis should be made by a healthcare professional based on a comprehensive examination and additional clinical information. | To determine whether the patient has glaucoma based on the fundus image, we need to look for signs commonly associated with this condition. Here are some key features to consider:....

Based on the visible features in this fundus image, there are no overt signs of glaucoma such as increased cupping, optic disc changes, or nerve fiber layer loss. However, a definitive diagnosis of glaucoma requires comprehensive evaluation, including intraocular pressure measurement, optic nerve head assessment, and visual field testing. In conclusion, the fundus image does not provide sufficient evidence to diagnose glaucoma. A comprehensive eye examination is necessary for an accurate diagnosis. |
| **MedAgent-Pro** | According to **relevant documents**, the  key indicators for diagnosing glaucoma from fundus  images are as follows:

 According to the **segmentation  result** of optic cup and disc,  the **vCDR is 0.612**, higher than the  risk thresold 0.6.

 The thickness of the patient's neuroretinal rim **deviates from the normal ISNT rule**, with a much thinner temporal margin.

The patient have **optic disc hemorrhage** with associated cup-to-disc and retinal pigmentation.

The image shows an clear optic disc boundary, which related to **peripapillary atrophy**.

Based on above findings, the patient is very like to have glaucoma. | According to **relevant documents**, the  key indicators for diagnosing glaucoma from fundus  images are as follows:

 According to the **segmentation  result** of optic cup and disc,  the **vCDR is 0.699**, higher than the  risk thresold 0.6.

 The thickness of the patient's neuroretinal rim **deviates from the normal ISNT rule**, with a much thinner temporal margin.

The patient have **optic disc hemorrhage** with associated cup-to-disc and retinal pigmentation.

The image shows an clear optic disc boundary, which related to **peripapillary atrophy**.

Based on above findings, the patient is very like to have glaucoma. |

Figure 11: Qualitative comparison on glaucoma diagnosis with general VLMs. Green texts highlight the correct conclusion made by MedAgent-Pro.

| | Original Image | |
|---|---|---|
| **Original Image** |  |  |
| **GPT-4o** | Cardiomegaly, or enlargement of the heart, is typically assessed on a posterior-anterior (PA) chest X-ray by measuring the cardiothoracic ratio (CTR) — the ratio of the cardiac width to the thoracic width. A CTR greater than 50% on a PA view is generally considered indicative of cardiomegaly. The heart size in the second image appears within normal limits. The width of the heart is less than half the width of the thoracic cavity, suggesting a normal cardiothoracic ratio. The patient does not appear to have cardiomegaly. | This chest X-ray shows a complex clinical picture with multiple devices and findings, but there does not appear to be an obvious acute bony fracture visible in the scanned region, including the clavicles, ribs, or spine at the level shown. 

 No obvious fracture visible on this image, but further clinical correlation and possibly advanced imaging are recommended to confirm or exclude any subtle fractures. |
| **Janus-Pro-7B** | Based on the given images, it is not possible for me to definitively determine if the patient has cardiomegaly. Cardiomegaly is typically assessed by comparing the size of the heart to the thoracic cavity on an X-ray, but these images only show the heart's location and size relative to the ribs and spine. Additional imaging or clinical evaluation would be needed to make a conclusive diagnosis. | Based on the provided X-ray image, there is no apparent fracture visible in the chest area. The bones appear to be intact and aligned correctly, with no visible signs of breakage or dislocation. |
| **LLaVA-Med** | Yes, the chest X-ray shows cardiomegaly, which is an enlargement of the heart. | Yes, the patient appears to have a Fracture, as indicated by the chest X-ray. |
| **Qwen2.5-7B-VL** | Based on the provided chest X-rays, it is not possible to definitively diagnose cardiomegaly (enlargement of the heart) without a detailed radiological interpretation by a qualified radiologist. However, there are some observations that can be made: 

 - The heart appears to be relatively large compared to the thoracic cavity. 

 - There may be some blunting of the costophrenic angles, which could suggest pleural effusion or other conditions. | The image you provided is a chest X-ray, which is used to visualize the structures of the chest, including the bones, lungs, and heart. In this particular X-ray, there are no obvious signs of a fracture. The ribs appear intact, and there are no visible fractures or breaks in the bony structures. 

 However, it's important to note that a single X-ray may not always be sufficient to diagnose a fracture, especially if the fracture is subtle or if the patient has had a recent trauma. If there is suspicion of a fracture, further imaging such as a CT scan might be necessary for a more detailed evaluation. |
| **InternVL-2.5-8B** | The image shows an X-ray of the chest. To determine if the patient has cardiomegaly, one would typically look for an enlarged cardiac silhouette, which may indicate an abnormal increase in heart size. 

 From the image, the heart appears to be enlarged relative to the thoracic cavity. This suggests that the patient may have cardiomegaly. However, a definitive diagnosis should be made by a medical professional, as additional information and context are necessary for an accurate assessment. | To determine whether the patient has glaucoma based on the fundus image, we need to look for signs commonly associated with this condition. Here are some key features to consider:.... 

 Based on the visible features in this fundus image, there are no overt signs of glaucoma such as increased cupping, optic disc changes, or nerve fiber layer loss. However, a definitive diagnosis of glaucoma requires comprehensive evaluation, including intraocular pressure measurement, optic nerve head assessment, and visual field testing. In conclusion, the fundus image does not provide sufficient evidence to diagnose glaucoma. A comprehensive eye examination is necessary for an accurate diagnosis. |
| **MedAgent-Pro** | Cardiomegaly, or enlargement of the heart, is typically assessed on a posterior-anterior (PA) chest X-ray by measuring the cardiothoracic ratio (CTR). CTR is he ratio of the cardiac width to the thoracic width. A CTR greater than 50% on a PA view is generally considered indicative of cardiomegaly. 

  The heart is located in the image below, the red box indicates the heart boundary. 

  The thoracic cavity is located in the image below, the red box indicates the thoracic boundary. 

 According to the result, the CTR is calculated as the ratio of the maximum horizontal cardiac diameter to the maximum horizontal thoracic diameter, which is **0.59** in this case. Judging from the above information, the patient is likely to have atelectasis. | Common radiographic findings of fractures in the thoracic vertebrae, ribs, clavicles, and sternum include: Cortical break: Disruption or alteration of the continuity or sharpness of the cortical bone lineAngulation or displacement: Local angulation or misalignment of bone fragments. Overlapping bone fragments, or abnormal changes in bone density, such as thickening, thinning, or irregularity of the cortex. 

  The thoracic vertebrae are located in the image.Juding from the results, the thoracic vertebrae appears relatively normal. 

  The ribs are located in the image below. Juding from the results, **old left-sided rib fractures** are present in the image. 

  The clavicles are located in the image below. Juding from the results, the clavicles appears relatively normal. 

 Judging from the above information, the patient is likely to have fracture. |

Figure 12: Qualitative comparison on chest X-ray diagnosis with general VLMs. Green texts highlight the correct conclusion made by MedAgent-Pro.

