# OpenReview forum: "MedAgent-Pro: Towards Evidence-based Multi-modal Medical Diagnosis via Reasoning Agentic Workflow"
_ICLR.cc/2026/Conference — ICLR 2026 Poster_

### Official Review · Reviewer_ACbS · 2025-10-27

**Soundness:** 3
**Presentation:** 4
**Contribution:** 4
**Rating:** 8
**Confidence:** 4

**Summary:**

This paper introduces **MedAgent-Pro**, an agent-based system designed for multi-modal medical diagnosis. The authors identify a key limitation in current Vision-Language Models (VLMs) and agentic systems: they tend to provide direct, empirical-driven answers without the step-by-step, evidence-based reasoning that is fundamental to real-world clinical practice.

To address this gap, MedAgent-Pro proposes a hierarchical workflow that mimics clinical procedure. This workflow is divided into two main stages: **(i)** Disease-Level Planning: A Retrieval-Augmented Generation (RAG) agent accesses a medical knowledge base to formulate a standardized, multi-step diagnostic plan based on established clinical guidelines. **(ii)** Patient-Level Reasoning: The agent executes this plan step-by-step, analyzing the specific patient's multi-modal data. This execution phase leverages a toolbox of "professional tools," such as segmentation models, to perform quantitative analysis (e.g., calculating the cup-to-disc ratio) alongside qualitative assessments.

The authors present an extensive evaluation across more than 10 imaging modalities and 50 diseases, claiming that MedAgent-Pro substantially outperforms general VLMs like GPT-4o, other medical agent systems, and task-specific expert models. The system's alignment with clinical workflows is further supported by positive evaluations from clinical experts. Overall, this is a fairly strong paper.

**Strengths:**

### **Originality and Significance:**
The most significant strength is the novel agentic paradigm itself. By formalizing the clinical diagnostic process into a hierarchical workflow (plan-via-RAG, execute-via-tools), the paper moves the field from "black-box" pattern recognition towards auditable, evidence-based reasoning. This directly addresses the critical needs of reliability, safety, and trustworthiness for medical AI.


### **Problem Formulation:**
The paper does an excellent job of motivating the work, clearly articulating the gap between current VLM capabilities (which provide "empirical-driven conclusions") and the needs of clinical practice (which demands "quantitative analysis" and structured reasoning).


### **Experimental Breadth:**
The evaluation is a major strength. The authors test their system on a very wide array of tasks, covering over 10 imaging modalities, 20+ anatomies, and 50+ diseases, providing strong evidence for the generalizability of the framework.


### **Clinician Evaluation:**
The inclusion of a qualitative evaluation by clinical experts is a crucial addition. Reporting that clinicians rated MedAgent-Pro's outputs higher on dimensions like "reasoning coherence" and "clinical reliability" provides strong, practical validation of the paper's central thesis.

**Weaknesses:**

### **Inadequate Contextualization with Prior Work:**

The paper claims that current medical agentic systems "simply glued all tools together". This may be an oversimplification that ignores recent, highly relevant work. The authors should cite and differentiate their approach from other multi-modal medical agents like "SMR-agents" (Wang et al., [1]) and "AURA" (Fathi et al., [2]), which also appear to integrate multi-modal reasoning and tool use in a sophisticated manner. A more nuanced comparison in the Related Work section is needed to clearly establish this paper's specific contributions over the state-of-the-art.

### **The "ToolBox" is Underspecified:**
The system's performance is critically dependent on its "Available ToolBox". The paper mentions "visual/coding models" and gives MedSAM as an example, but a comprehensive list of all tools, their specific capabilities, and their individual performance is not provided. Without this, it's impossible to assess the agent workflow independently of the tools.

### **"Evidence-Based Reflection" is Unclear:**
 This mechanism is highlighted as a key contribution but is operationally vague. How does the agent "evaluate the reliability" of a step's output? Is this a heuristic, a learned classifier, or a VLM self-critique prompt? The "Evi. Reasoning" block in Figure 2 is abstract. This core mechanism needs to be formalized.


### **Conceptual Limitation to Sequential Reasoning:**
The proposed methodology is inherently bounded to sequential reasoning steps. The RAG agent generates a linear plan, and the execution agent follows it. The paper does not discuss or explore scenarios that might require parallel exploration of different diagnostic paths or more complex, branching decision-making. This is a notable limitation, as real-world diagnostics can often be non-linear.

### **Confusing Mathematical Notation:**
The mathematical expressions in Section 3.2, such as representing inputs and outputs as key-value pairs (e.g., $\mathcal{P}i:r_{i}=\langle k_{r_{i}},v_{r_{i}}\rangle$), add a layer of formalism that is arguably unnecessary and confusing. A-plainer textual description of how data is passed and transformed between steps would significantly improve clarity and accessibility.

### **RAG Knowledge Base:**
The paper relies on a "Medical Knowledge Base" sourced from MedlinePlus. The scale, and curation of this knowledge base, as well as the indexing and retrieval process, are mentioned but could be detailed further. The quality of the RAG-generated plan is a critical bottleneck that is not explicitly evaluated.

[1] Wang et al., "SMR-agents: Synergistic medical reasoning agents for zero-shot medical visual question answering with MLLMs"

[2] Fathi et al., "AURA: A Multi-modal Medical Agent for Understanding, Reasoning and Annotation"

**Questions:**

See "Weaknesses" for questions. **Note** that, if the related works section and other problems in Weaknesses are not addressed the rating can be subject to deduction.

---

> ### Author Response · Authors · 2025-11-21
> **Reply to Reviewer ACbS (Part 1)**
>
> **W1:**
> Thanks for the valuable suggestion. We acknowledge that our original statement may be overly broad, and recent works such as *SMR-agents* and *AURA* were inadvertently omitted due to their very recent publication.
>
> To clarify, *SMR-agents* focuses on zero-shot medical VQA, and *AURA* emphasizes multimodal tool orchestration for medical image understanding. In contrast, the distinctive contribution of *MedAgent-Pro* lies in two aspects:
>
> 1. **Knowledge-based planning** that retrieves medical knowledge to organize tools for complex diagnosis that align with clinical criteria;
>
> 2. **Evidence-based reasoning** that incorporats a self-reflection mechanism that validates each reasoning step, thereby enhancing the reliability of the diagnostic process.
>
> We have **included** both citations and revised the **Introduction** and **Related Work** section to explicitly clarify these distinctions.
>
> **W2:**
> We appreciate the suggestion for a more detailed description of the available toolset. To address this concern, we have now added a comprehensive tool description to **Appendix E.5**. The table below lists all tools used in MedAgent-Pro, including their usage, associated task, implementation, and whether additional training was performed.
>
> | Usage                  | Task              | Tool / Model        | If Trained |
> | ---------------------- | ----------------- | ------------------- | ---------- |
> | Segmentation           | Glaucoma          | Medical SAM-Adapter | No         |
> |                        | Heart Disease     | Medical SAM-Adapter | Yes        |
> |                        | MIMIC-CXR         | MedSAM              | No         |
> |                        | NEJM Chest X-ray  | MedSAM              | No         |
> |                        | NEJM CT & MRI     | MedSAM              | No         |
> |                        | NEJM Cell Imaging | Cellpose            | No         |
> | Grounding              | MIMIC-CXR         | Maira-2             | No         |
> | Coding  | All tasks         | GitHub Copilot      | No         |
>
> As shown, **only** the glaucoma and heart-disease datasets contain ground-truth masks, enabling us to quantitatively evaluate the performance of corresponding tools. Other tasks **do not** provide annotations for quantitative assessment, and the tools are used in a zero-shot manner, and we refer readers to their original papers for their general performance.
>
> | Segmentation Task | Dice | IoU  |
> | ----------------- | ---- | ---- |
> | Optic Disc        | 0.97 | 0.90 |
> | Optic Cup         | 0.87 | 0.80 |
> | Myocardium        | 0.87 | 0.77 |
> | LV Cavity         | 0.78 | 0.70 |
>
> Notably, as demonstrated in **Figure 5** of the main text, our workflow is robust to the accuracy of individual tools, indicating that our performance does not depend on tool-specific metrics. We have included this table in **Appendix E.5** of the revised manuscript to provide a detailed description of the MedAgent-Pro toolBox.
>
> **W3:**
> Thanks for the question. *Evidence-Based Reflection* is a zero-shot, prompt-based reliability check process executed by the VLM. For each step output $r_i,$ the VLM assesses:
>
> 1. whether the input image quality allows the required action,
> 2. whether visual-tool outputs (masks/boxes) have reasonable size, shape, and anatomical location,
> 3. whether numerical results fall within valid ranges defined by the guideline (e.g., $CDR < 1$ in glaucoma diagnosis), and
> 4. whether qualitative descriptions are supported by clear visual clues and not expressed with uncertainty.
>
> Based on this assessment, the state function returns whether the result is reliable.
> No learned classifier is used, preserving zero-shot fairness.
>
> A more detailed operational description and examples have been added in **Appendix E.4** of the revised manuscript, and referenced in **Section 3.3** for clarity.

---

> ### Author Response · Authors · 2025-11-21
> **Reply to Reviewer ACbS (Part 2)**
>
> **W4:**
> We agree that real-world clinical diagnostics may involve non-linear reasoning. We clarify the following:
>
> 1. Clinicians in practice typically narrow down to *a small set of suspected conditions* before further differentiating among them [1]. MedAgent-Pro generally follows this realistic workflow, simplifies the open-world diagnosis into binary or MCQ tasks.
>
> 2. Our diagnostic workflow is *not strictly linear*. For a given disease, MedAgent-Pro evaluates multiple indicators in parallel, where each indicator is independently analyzed before aggregation. This parallel, indicator-level reasoning shows that the framework naturally supports extensions toward branching diagnostic exploration.
>
> 3. In the MCQ setting for the NEJM dataset, MedAgent-Pro also performs *parallel multi-path reasoning*. The system constructs diagnostic plans for each candidate option, analyzes each option separately, and then lets the VLM integrate indicator-level results to select the most plausible answer. This demonstrates that MedAgent-Pro already goes beyond a single linear chain.
>
> We have made further clarification clarified in the revised **Appendix E.1**, explaining how the workflow handles MCQ problems parallelly.
>
> [1] Falik, Ruth. "Symptom to Diagnosis: An Evidence-Based Guide." JAMA: The Journal of the American Medical Association 295.22 (2006): 2667-2668.
>
> **W5:**
> Thanks for pointing out the problem. Following the reviewer’s suggestion, we have simplified the mathematical notation in **Section 3.2–3.3** by removing the key–value pair formalism and replacing it with a clearer expression (e.g., we now define the plan as $(P_i:\ r_i = a_i(o_i),\ a_i \in \mathcal{A})$) to improve readability. These revisions enhance clarity and accessibility of the formulation. Please refer to **p.4 and p.5** in the updated version. Thanks again for the suggestions!
>
> **W6:**
> We appreciate the thoughtful comment. To clarify:
>
> 1. MedlinePlus is an NIH/NLM-backed, publicly accessible health information resource containing over 1,000 disease and wellness entries and more than 4,000 expert-reviewed articles. Its structured topic hierarchy and curated metadata naturally support reliable indexing, allowing our system to efficiently identify clinically relevant content for a given query.
>
> 2. As illustrated in *Figure 3*, our retrieval pipeline conduct following steps:
> - Each MedlinePlus article is first assigned a one-sentence summary with its metadata, which serves as a lightweight index for coarse filtering.
> - Given a queried condition, the RAG agent uses this index to obtain a query-relevant subset of documents.
> - These documents are then segmented into 300-token chunks, embedded with PubMedBERT, and stored in a FAISS/LangChain vector index. Top-5 similarity retrieval is performed over these embeddings, and the retrieved chunks are summarized by the VLM to construct the final diagnostic plan.
>
> 3. Regarding the quality of the RAG-generated plan, firstly, in **Figure 6** of the revision version, we evaluate how the generated plan aligns with real clinical workflows, where chest X-ray tasks show a clear positive correlation.
> Secondly, we demonstrate robustness of the system to variations in the RAG process. As shown in **Appendix B.2**, diagnostic performance remains stable across different retrieval backends and knowledge sources.
>
> We have revised the description of the RAG knowledge base and retrieval pipeline in **p.4** of the main paper for clarity and reproducibility. Thanks again for the suggestion!

---

> > ### Comment · Reviewer_ACbS · 2025-11-28
> >
> > I appreciate your detailed response. Most of my concerns are addressed and answered. I will keep my score as is.

---

> > > ### Author Response · Authors · 2025-11-29
> > > **Official Reply by Authors**
> > >
> > > Thanks for the insightful review and the constructive feedback after reading the rebuttal. We are glad that our clarifications addressed your questions. Thanks again for your effort.

---

### Official Review · Reviewer_HerW · 2025-10-27

**Soundness:** 3
**Presentation:** 3
**Contribution:** 3
**Rating:** 6
**Confidence:** 4

**Summary:**

This work explores reasoning agentic system of medical diagnosis. By disease-level standardized plan generation and patient-level personalized step-by-step reasoning, MedAgent-Pro presents modern diagnosis principles via a hierarchical diagnostic workflow. They leverages professional tools such as visual models to take various actions to analyze multi-modal input, and performs evidence-based reflection. Extensive experiments across a wide range of anatomical regions, imaging modalities, and diseases are conducted.

**Strengths:**

1.	Evaluations across 10+ imaging modalities, 20+ anatomies, and 50+ diseases validate its universality.
2.	The proposed system fully leverages existing models in their toolbox to provide clinical evidence.
3.	The proposed model is trustworthy because of more visual cues and indicators compared to vanilla VLMs.
4.	Comprehensive ablations further validate the effectiveness of major components in the agentic system.

**Weaknesses:**

1.	The author proposed evidence-based system. However, the proposed system is more like an engineering product, which lack of architecture innovation and similar to other agentic systems[1,2].
2.	The author should provide analysis in view of visual cues compared to medical world model[3], which was also designed for treatment planning. Is the proposed system providing any visual evidence?
3.	Which part of MedAgent-pro is finetuned? Is the backbone GPT-4o finetuned on downstream data? What about the compared methods? The fairness should be ensured.
4.	In Evidence-based Reflection, is it possible that s_i is always continue, resulting in unexpected dead loop?
5.	Clinical guidelines may be inconsistent across different areas and hospitals. How can the proposed system overcome such inconsistency?

[1] Zhu Y, He Z, Hu H, et al. MedAgentBoard: Benchmarking Multi-Agent Collaboration with Conventional Methods for Diverse Medical Tasks[J]. arXiv preprint arXiv:2505.12371, 2025.
[2] Zhu Y, Qi Y, Wang Z, et al. HealthFlow: A Self-Evolving AI Agent with Meta Planning for Autonomous Healthcare Research[J]. arXiv preprint arXiv:2508.02621, 2025.
[3] Yang Y, Wang Z Y, Liu Q, et al. Medical world model: Generative simulation of tumor evolution for treatment planning[J]. arXiv preprint arXiv:2506.02327, 2025.

**Questions:**

Please see weaknesses

---

> ### Author Response · Authors · 2025-11-21
> **Reply to Reviewer HerW (Part 1)**
>
> **W1:**
> Thank you for your comment. While some components of our system have been proposed before, our framework focus on presents an **carefully-designed architecture** to provide clinically-grounded reasoning for complex diagnosis rather than a collection of independent modules.
>
> Regarding the cited works, *MedAgentBoard* is a benchmarking framework rather than a technical innovation. It provides valuable insights by showing that multi-agent collaboration must be justified by task-specific and *evidence-driven* designs. This directly motivates the design of MedAgent-Pro: we build a structured workflow explicitly grounded in medical guidelines and visual evidence. *HealthFlow* focuses on text-centric meta-planning for autonomous healthcare research without *integrating medical imaging*. MedAgent-Pro specifically fills this gap through the integration of quantitative visual tools.
>
> In summary, MedAgent-Pro differs from these systems through
> 1. Medically grounded disease-level planning to orchestrate different agents,
> 2. Qualitative and quantitative reasoning with integrated visual tools, and
> 3. Step-wise verification of intermediate results.
> These components enable structured, interpretable, and clinical-aligned diagnostic workflows that are not present in existing agentic systems.
>
> To more clearly position these distinctions, we have revised the **Introduction** and **Related Work** sections to *include discussion of MedAgentBoard and HealthFlow*, and to highlight how MedAgent-Pro contributes a clinically oriented workflow.
>
> **W2:**  Thanks for bringing up the Medical World Model (MeWM).
> Although both MeWM and MedAgent-Pro incorporate *visual evidence*, they serve **different purposes** and obtain such evidence through different models:
>
> *Technically*, MedAgent-Pro leverages **existing visual inputs** and derives evidence through specialized tools such as segmentation (e.g., optic disc/cup extraction) or grounding models (e.g., locating effusions in chest X-rays). Meanwhile, MeWM produces visual evidence by **generating** synthetic disease-progressive images using a generative model.
>
> *Motivationally*, MedAgent-Pro uses visual evidence to support **structured clinical reasoning**, helping the system evaluate diagnostic indicators.
> MeWM, however, uses generated visual trajectories to **simulate treatment outcomes**, aiming to guide therapeutic decision-making rather than diagnostic reasoning.
>
> In summary, while both systems incorporate visual evidence, they differ in task objective, how visual evidence is obtained, and how it is used within the reasoning process. We have added this clarification to **Appendix C.2** of the revised manuscript.
>
>
> **W3:** We thank the reviewer for raising this important point.
> To ensure fairness, all backbone VLMs used in MedAgent-Pro and comparison methods (e.g., GPT-4o, Qwen-VL, InternVL) are kept zero-shot without any finetuning on all tasks. The RAG module also adopts existing solutions without additional training. Only a few specialized tool-agents (such as the several segmentation models in the glaucoma task) are trained for *domain-specific* processing. We have added detailed description of toolset in **Appendix E.5**.
>
> In additional, as shown in **Appendix B.4** (as well as **Table 6** in revision manuscript), when we allow other baselines to access the same toolset, MedAgent-Pro still achieves the best performance, demonstrating that its advantage stems from the *hierarchical agentic workflow* rather than extra training, which is also acknowledged by *Reviewer nJkL*. We have clarified these details in the revision, including updates in **Appendix B.4** and related methodological descriptions.

---

> ### Author Response · Authors · 2025-11-21
> **Reply to Reviewer HerW (Part 2)**
>
> **W4:** We fully understand the reviewer’s concern. In practice, MedAgent-Pro avoids such cases through two safeguards:
> 1. During the *knowledge-based reasoning* stage, the system formulates disease-specific diagnostic plans based on retrieved medical knowledge and the *available toolset*, ensuring that each planned step is **executable**. This prevents unresolvable loops from arising.
> 2. During runtime, we also set a strict **upper limit** on reasoning rounds (10). If this limit is reached, the corresponding branch is terminated and excluded from the final decision-making, which guarantees stability. In all experiments, this boundary condition was almost never triggered.
>
> We have added a detailed explanation of this mechanism in **Appendix E.4** of the revised manuscript.
>
> **W5:** We fully acknowledge the inconsistency in real-world medical practice. We would like to clarify that MedAgent-Pro is designed as a **guideline-agnostic** and **modular** framework, rather than depending on any specific guideline source.
>
> 1. The *knowledge-based planning* primarily focuses on determining key diagnostic indicators (e.g., LVEF, CDR). These indicators are widely acknowledged and **highly consistent** across major guidelines [1]. Variation typically arises in the relative emphasis placed on different indicators by individual areas or hospitals, meaning that the diagnostic plan and the patient-level reasoning remains stable.
> 2. MedAgent-Pro is designed as a flexible workflow that allows different hospitalsto *incorporate their own guidelines*. Replacing the knowledge source requires no change to the overall workflow, enabling a customized application.
> 3. In the US, hospitals commonly rely on ACC/AHA guidelines for heart-disease diagnosis. To support our claim, we added an additional experiment in **Appendix B.2**, comparing different RAG sources:
>
> | RAG Source            | Heart Disease bAcc | Heart Disease F1 |
> | --------------------- | ------------------ | ---------------- |
> | MedlinePlus (Default) | 77.8               | 72.3             |
> | ACC/AHA Guidelines    | **78.9**           | **74.0**         |
>
> Our additional experiment shows that substituting MedlinePlus with ACC/AHA yields nearly identical diagnostic performance. Combined with other results in **Appendix B.2**, this demonstrates that MedAgent-Pro is robust to reasonable variations in guideline sources.
> We sincerely thank the reviewer for this insightful suggestion on hospital-specific guidelines or private RAG sources, and we have added this discussion to **Appendix B.2**, as such customization further enhances the practical clinical relevance of MedAgent-Pro.
>
> [1] Mainz, Jan. "Defining and classifying clinical indicators for quality improvement." International journal for quality in health care 15.6 (2003): 523-530.

---

### Official Review · Reviewer_nJkL · 2025-10-31

**Soundness:** 3
**Presentation:** 3
**Contribution:** 3
**Rating:** 6
**Confidence:** 4

**Summary:**

The paper introduces MedAgent-Pro, an agentic reasoning workflow designed to emulate evidence-based clinical diagnosis. The system operates on a hierarchical structure, using planning, RAG and reasoning agents to ensure reliability.

**Strengths:**

The primary strength of this work is its alignment with medical practice, from an empirical one-hop VQA to a structured and evidence-based reasoning process.

**Weaknesses:**

My major comments are:

1. The system's effectiveness, particularly its quantitative analysis which drives the performance gains, relies on the availability of specialized visual tools for a given task. The workflow's performance for diseases or modalities lacking such robust, pre-existing tools is less clear. While the NEJM results without tools are still strong, they lack the key quantitative grounding that differentiates the method.
2. The VLM  remains a single point of failure for several critical steps. The VLM is solely responsible for generating the entire diagnostic plan from guidelines, performing all qualitative analysis, executing the evidence-based reflection, and assigning the final risk-based weights. Any hallucination or error in these steps could compromise the entire workflow.
3. The main comparison tables compare MedAgent-Pro (with tools) against general VLMs (without tools). This confounds the benefit of the agentic workflow with the benefit of simply having tool access. A more rigorous comparison, presented only in Appendix B.4, shows that while MedAgent-Pro still wins, giving baselines tool access does significantly close the gap. This stronger, more fair comparison should have been centered in the main paper.
4. The evaluation on the MITEA dataset (heart disease) was simplified from a 7-class problem to a binary classification task (healthy vs. heart disease) due to limited samples per category. This simplification is a significant departure from a realistic clinical scenario, where differentiating between various heart conditions (e.g., amyloidosis vs. hypertrophy) is the critical diagnostic task.
5. The RAG agent's knowledge base is MedlinePlus. However, clinical-grade diagnosis typically relies on more complex professional guidelines from medical societies (e.g., ACC/AHA, etc.).

**Questions:**

Please see the weaknesses above

---

> ### Author Response · Authors · 2025-11-21
> **Reply to Reviewer nJkL (Part 1)**
>
> **W1:** Thanks for the comment. We clarify that visual tools are an important component of our framework, but not a must:
>
> 1. The core contribution of MedAgent-Pro is providing a *structured, medical-grounded reasoning framework*. The workflow decomposes a disease diagnosis into step-by-step analysis over a set of indicators, performs *evidence-based reasoning* for each indicator, and aggregates these findings into a final decision. While integrating specialized visual tools can enhance individual indicator assessments, the workflow itself **does not** rely on specific tools.
>
> 2. As you mentioned, results on the NEJM dataset further demonstrate this property. Even without visual tools, MedAgent-Pro still produces structured reasoning with indicator-level qualitative analysis that leads to reliable conclusions. This contrasts with role-playing based agent systems, which often lack structured indicator analysis and medical guideline guidance.
>
> 3. MedAgent-Pro is also inherently **extensible**. Users can plug in their *customized tools* or directly provide pre-measured indicators to support diagnosis. These design choices ensure that MedAgent-Pro maintains clear advantages over existing methods through its carefully structured workflow.
>
> **W2:** We appreciate the reviewer’s insightful comment.
> While the VLM acts as the central coordinator in MedAgent-Pro, it is *not* a single point of failure. The system incorporates multiple safeguards to ensure robustness and reliability.
>
> 1. During the planning stage, MedAgent-Pro retrieves authoritative medical guidelines through the RAG agent and verifies each diagnostic step with the available toolset. This ensures that the generated plans are clinically reliable and executable, thereby minimizing hallucination at the planning stage.
>
> 2. VLM may still exhibit hallucination during qualitative analysis. To mitigate this, the RAG module provides guideline-aligned context for every reasoning step, grounding each decision in domain knowledge. In addition, (i) specialized qualitative tools are limited across diverse medical scenarios, and (ii) as shown in **Table 5** of the main paper, such tools provide only marginal gains, indicating that our workflow design, rather than model choice, drives reliability.
>
> 3. The evidence-based reflection mechanism (**Sec. 3.3**) dynamically evaluates the reliability of each reasoning step. Unreliable predictions trigger a `Terminate` state rather than continued propagation, preventing error accumulation. This mechanism ensures each step remains aligned with the guideline and effectively minimizes risks associated with VLM hallucination.
>
> 4. The risk-based weight for each indicator is assigned based on clinical guidelines, and unreliable indicators are excluded from decision-making Although training task-specific weights could further improve performance, we intentionally keep weights rule-based to maintain **zero-shot fairness**.
>
> Thanks for the suggestion again, and we have revised details in the main paper to prompte clarity against the concern.
>
> **W3:** We appreciate the reviewer’s thoughtful suggestion. As noted, **Appendix B.4** shows that even when baselines are equipped with the same toolset, MedAgent-Pro still outperforms them, confirming that the performance gain originates from our hierarchical agentic workflow rather than from tool usage alone. In addition, **Appendix B.1** demonstrates consistent improvements across different backbone VLMs, further underscoring the robustness of our design.
>
> To improve clarity and fairness, the **revision PDF** now moves the **Appendix B.4** results into the main paper, replacing the less critical **original Figure 6** (now relocated to the **Appendix B.5**). These updates are reflected on **page 8** of the revised version. Thanks again for the suggestion, which ensures the main paper directly presents the fair, tool-controlled comparisons and highlights the robustness of our framework.

---

> ### Author Response · Authors · 2025-11-21
> **Reply to Reviewer nJkL (Part 2)**
>
> **W4:**
> Thank you for the insightful comment. The MITEA dataset is highly imbalanced across its 7 categories (6 abnormalities + 1 healthy), as shown in the table below. This imbalance makes multi-class evaluation less reliable.
> To address your concern, we have additionally included the full multi-class results in **Appendix A.6** as also shown below:
>
>
> | Method         | Avg  | Amyloidosis | Aortic Regurgitation | Cardiomyopathy Dilated | Cardiomyopathy Hypertrophic | Hypertrophy | Transplant |
> |----------------|------|-------------|----------------|------------|------------------|-------------|-----------|
> | Cases          |  /   | 12          | 10             | 6          | 8                | 14          | 2         |
> | GPT-4o         | 56.8 | 65.6        | 50.6           | 58.6       | 60.5             | 52.4        | 53.0      |
> | Janus-Pro-7B   | 49.3 | 57.6        | 35.8           | 49.1       | 53.9             | 64.6        | 34.5      |
> | LLaVA-Med      | 51.6 | 50.0        | 54.1           | 49.6       | 55.9             | 50.0        | 50.0      |
> | Qwen2.5-7B-VL  | 49.4 | 48.8        | 50.0           | 49.6       | 48.9             | 50.0        | 48.9      |
> | InternVL2.5-8B | 52.5 | 54.9        | 51.8           | 62.6       | 48.1             | 48.5        | 48.9      |
> | MedAgent-Pro   | **82.8** | **93.0** | **80.2** | **87.4** | **85.1** | **80.7** | **70.5** |
>
> As shown, MedAgent-Pro achieves **substantially higher performance** when diagnosing the more specific heart-disease subtypes, consistently outperforming all VLM baselines. The improvement illustates that a clearer disease query allows MedAgent-Pro to focus on more direct and clinically meaningful indicators (e.g., ventricular wall thickening for amyloidosis), enabling more accurate and grounded reasoning. Thanks again for this valuable suggestion!
>
> **W5:**
> Thanks for the thoughtful comment. Below we provide additional explanation regarding the RAG module:
>
> 1. MedlinePlus is expert-reviewed, publicly accessible, and organized into a well-structured hierarchy of disease topics. Since our goal is to design a *generalizable and flexible* framework, MedlinePlus naturally serves as a widely applicable knowledge source for implementing the RAG module across different clinical scenarios. We did not integrate more professional RAG database for each task since its not the research focus.
>
> 2. Importantly, the RAG agent **does not** rely on any MedlinePlus-specific formatting. Specialized guidelines such as ACC/AHA documents can be directly incorporated, and real hospitals may plug in their own private guideline repositories. The workflow remains fully compatible without modification.
>
> 3. To further address your concern, we additionally replaced MedlinePlus with **ACC/AHA guideline documents** for the Heart Disease task. The results are as follows:
>
> | RAG Source            | Heart Disease bAcc | Heart Disease F1 |
> |-----------------------|--------------------|------------------|
> | MedlinePlus (Default) | 77.8               | 72.3             |
> | ACC/AHA Guidelines    | **78.9**           | **74.0**         |
>
> Using a more specialized knowledge base slightly improves performance. We have added this experiment in the revised **Appendix B.2**. Thanks again for the helpful suggestion. Incorporating professional or hospital-specific guidelines further enhances the practical adaptability of MedAgent-Pro.

---

> ### Comment · Reviewer_nJkL · 2025-11-26
>
> Thanks to the authors for their response. Most of my questions have been addressed. I'll keep my score.

---

> > ### Author Response · Authors · 2025-11-27
> > **Offical Comment by Authors**
> >
> > We sincerely thank you for reading our rebuttal and for the constructive feedback. We are glad that our clarifications addressed your questions. We would also be happy to engage in further discussion for any additional problems.

---

### Official Review · Reviewer_roDJ · 2025-11-02

**Soundness:** 3
**Presentation:** 3
**Contribution:** 2
**Rating:** 2
**Confidence:** 5

**Summary:**

This paper introduces MedAgent-Pro, an agentic hierarchical workflow system for evidence-based, multi-modal medical diagnosis. The approach separates disease-level planning (standardized per guidelines using a retrieval-augmented generation agent) and patient-level stepwise reasoning empowered by professional tools (e.g., visual models, coding agents), integrating iterative reflection for reliability. MedAgent-Pro is benchmarked on extensive multi-modal datasets, showing performance gains over mainstream VLMs, agentic baselines, and expert models.

**Strengths:**

1. The hierarchical design effectively mirrors medical diagnostic principles, with disease-level planning driven by retrieval from guidelines and patient-level personalized analysis.
2. The RAG-based incorporation of external domain knowledge into the planning stage supports clinical transparency.

**Weaknesses:**

1. Although the hierarchical structuring and rigorous evidence focus are valuable, the architecture closely resembles recent work in multi-agent, RAG-augmented, and tool-integrated medical AI (e.g., MedAgents, MMedAgent, MDAgent, and others in Table 1). Moreover, the overall contribution of this work appears rather limited. Many of the claimed innovations rely on techniques that have already become widely adopted or established in the field. As such, the paper seems more like an integration or system-level implementation of several existing popular paradigms, rather than presenting a novel method.

2. Can the authors clarify and mathematically specify the state assessment function $\phi$ used during patient-level reasoning? How is output reliability judged, is it a learned model, rule-based, or probabilistically estimated?

3. How are risk-based weights $\mathcal{W}$ in risk score computation (p.5) defined in practice? Are they derived from clinical guidelines directly, learned via validation, or optimized by the system?

4. For clinical expert evaluation, please elaborate on the number of raters, the randomization protocol, inter-rater agreement metrics (such as Cohen’s kappa or ICC), and whether any anchoring or exposure effects could have occurred.

**Questions:**

see Weaknesses

---

> ### Author Response · Authors · 2025-11-21
> **Reply to Reviewer roDJ (Part 1)**
>
> **W1:** We appreciate your insightful concern. While some of the techniques have appeared in recent works, we want to clarify that our motivation is not to propose new models, but rather to build a *flexible agentic workflow* for clinically grounded diagnosis. Our main contribution is offering a framework that organizes existing tools in a clinical-grounded way to perform long-chain diagnostic reasoning instead of *statically* using tools., which has not been explored in prior work. Specifically, the innovations of MedAgent-Pro include:
>
> 1. disease-level planning that retrieves and composes guideline-grounded diagnostic steps to dynamically integrate different tools;
> 2. patient-level reasoning that integrates qualitative and quantitative tools for structured indicators analysis;
> 3. a step-wise reflection mechanism that ensures intermediate outputs are clinically valid before proceeding.
>
> Together, these components form a structured and clinically aligned reasoning paradigm. These distinctions separate our framework from existing medical agentic systems such as *MMedAgent* or *MDAgent*, which primarily rely on a VLM’s internal knowledge without alignment to clinical protocols or without integrating visual evidence into reasoning. As you noted in **Table 1** of the main text, MedAgent-Pro outperforms these systems by a substantial margin across multiple tasks (e.g., **+33.6%** and **+20.6%** bAcc on glaucoma and heart disease), indicating that our clinically grounded workflow leads to fundamentally different outcomes compared to previous works.
>
> To better highlight these advantages, we have revised the **Introduction** and **Related Work** sections to position MedAgent-Pro as a system-level, clinically oriented workflow built upon existing paradigms, and to clarify its differences from prior methods.
>
> **W2:** Thanks for the suggestion on further formalizing the state-assessment function. In the revision, we explicitly clarify the function used in the evidence-based reflection as:
>
> $$
>  \phi: (r_i, o_i, G) \rightarrow \{{\texttt{Reliable},\ \texttt{Unreliable}}\},
> $$
>
> where $r_i$ is the output of step $P'_i$ and $G$ is the retrieved clinical guideline, which can be (i) a visual model output (e.g., segmentation mask or bounding boxes), (ii) a numerical result (e.g., the cup-to-disc ratio), or(iii) a qualitative analysis.
> Importantly, $\phi$ is *not* a learned model or classifier; it is implemented as a zero-shot evaluator executed by the baseline VLM to ensure a fair comparison with other baseline.  Specifically, the VLM judges the reliability of $r_i$ along following aspects:
> 1. The reflection prompt explicitly asks the VLM to judge the quality of the input image and the feasibility of performing the specified action (e.g., “segment optic disc” or “assess pleural effusion”). The VLM evaluates whether the image is blurry, occluded, cropped, low-resolution, or missing the relevant anatomical region.
> 2. If $r_i$ is the output of a visual tool, the VLM is prompted to assess whether the predicted region is anatomically plausible. The VLM examines (i) whether the mask or bounding box has a reasonable size (e.g., masks or boxes that cover nearly the entire image are clearly incorrect), (ii) whether the shape is reasonable (e.g., the mask contains large holes or appears fragmented), and (iii) whether the prediction is consistent with the clinical guideline
> (e.g., “the cup region should lie inside the disc region”).
> 3. When $r_i$ is a numerical value (e.g., vCDR ratio or LVEF estimate), $\phi$ checks whether the value lies within a physiologically valid range based on the guideline $G$ (e.g., the vCDR value should be less than 1).
> 4. When $r_i$ is a qualitative statement produced by the VLM or a grounding tool (e.g., “mild opacity in the right lower lobe”), the reflection prompt requires that such judgments be made only when clear visual or textual clues are present. If the VLM expresses uncertainty (e.g., “the abnormality is not clearly visible”), the result is considered unreliable.
>
> Based on this evaluation, $\phi$ returns whether the result is reliable. This rule-based design eliminates task-specific training and allows the evidence-based reflection mechanism to generalize across diverse medical scenarios without being constrained by *training data availability*. The detailed description has been added to **Appendix E.4** in the revised paper, and referenced in **Section 3.3** for improved clarity. Thanks again for the suggestion!

---

> ### Author Response · Authors · 2025-11-21
> **Reply to Reviewer roDJ (Part 2)**
>
> **W3:** Thanks for your valuable suggestion. We make the following explanation:
> 1. The risk-based weights are derived from the *clinical importance* of diagnostic indicators. During inference, the VLM receives both the guideline $\mathcal{G}$ obtained in knowledge-based planning and the final indicators $\mathcal{R}_{\text{final}}$ from evidence-based reasoning, and assigns relative weights according to guideline-defined medical significance. For example, key indicators such as LVEF reduction or CDR enlargement are clinically more decisive and therefore receive higher weights.
>
> 2. The weights $W$ used in the risk-based decision are **not trained or optimized** in any way. We intentionally adopt a **zero-shot workflow** without finetuning VLMs or introducing task-specific parameters to ensure fairness and avoid adding supervision that baselines do not have. While learning task-specific weights could yield small improvements, it would violate the zero-shot setting.
>
> 3. Although repeated runs may introduce slight variations in these weights, our analysis shows that such variations do not largely affect performance. As shown in **Appendix A.4**, MedAgent-Pro exhibits substantially lower variance compared to pure VLM baselines, demonstrating robustness to weight choices.
>
> In the revised version, we have improved the description in **p.5** to make the process clearer.
>
> **W4:** Thanks for raising this important point. We clarify our clinical expert evaluation protocol below:
>
> 1. Two licensed clinicians participated in the study: one ophthalmologist evaluated the glaucoma cases, and one thoracic surgeon evaluated the chest X-ray cases. Each rater assessed only cases within their clinical specialty, ensuring *domain-appropriate* judgment.
>
> 2. To mitigate anchoring or exposure effects, we used a randomized and blinded evaluation protocol. For each task, all model outputs (MedAgent-Pro and comparison methods) were anonymized, and both the case order and the ordering of outputs for each case were randomly shuffled for the clinician. This prevents bias from fixed sequencing or repeated exposure.
>
> 3. Since each task was evaluated by a single expert, inter-rater agreement metrics such as Cohen’s $\kappa$ or ICC are not applicable. Instead, we ensured reliability through strict randomization, blinding, and specialty-matched evaluation.
>
> These protocol details and the limitation of single-rater evaluation have been added to **Appendix E.3** in the revised version. Thanks again for prompting this clarification.

---

### Author Response · Authors · 2025-11-21
**General Replyl by Authors**

Dear Area Chairs and Reviewers,

We sincerely thank all reviewers for their time and thoughtful assessment of our submission. We are encouraged by the positive comments shared across the reviews, including t[clinical grounding] and [structured diagnostic workflow] by Reviewers **nJkL** and **ACbS**, [comprehensive evaluation] by Reviewers **HerW** and **ACbS**, [clarity of presentation] from Reviewers **roDJ** and **ACbS**. We greatly appreciate these acknowledgement of the motivation and contributions of our work.

Several concerns, especially from *Reviewer roDJ*, seem to arise from a different interpretation of our system’s design. We would like to clarify that our intention is not to introduce a new model, but rather to present a **flexible agentic workflow** for clinically grounded diagnosis. Our main contribution is offering a framework that organizes existing tools in a clinical-grounded way to perform long-chain diagnostic reasoning instead of statically using tools.
We also understand that some comments may reflect viewing our tasks as conventional VQA; however, our framework is designed for complex diagnostic scenarios that require *multi-step reasoning* and *tool-based quantitative analysis*, which go beyond typical VQA settings.
We have improved the **Introduction** and **Related Work** sections to clearly distinguish our approach from recent systems such as SMR-agents, AURA, MedAgentBoard (NeurIPS 2025), and HealthFlow, as highlighted by the reviewers.

We have also addressed all remaining concerns point-by-point, including those related to evidence-based reflection, implementation details, and the RAG knowledge base. The revised manuscript incorporates the following updates:
1. revisions to the **Introduction**, **Related Work**, and **Appendix C.2** to better articulate our motivation and clarify connections to existing systems;
2. refined descriptions of the RAG pipeline and workflow formulation on **p.4** and **p.5**;
3. reorganized toolset ablation results into **p.8** of the main text;
4. additional experimental results for the MITEA sub-tasks in **Appendix A.6**;
5. additional experiments using different RAG knowledge sources in **Appendix B.2**; and
6. detailed clarification of the toolset and evidence-based reflection mechanism in **Appendix E.4** and **Appendix E.5**.

We thank all reviewers again for their valuable feedback, have **highlighted all modifications in blue** in the revised manuscript, and welcome further discussion if needed.

Sincerely,

Authors of Submission 3594

---

### Author Response · Authors · 2025-12-01
**Summarization of Rebuttal**

Dear Area Chairs and Reviewers,

Thanks very much for your effort and time during this unusual review period. We understand that the changes in the AC assignment and review workflow have likely increased your burden. We are submitting this additional reply to briefly summarize how our rebuttal addresses the major concerns raised and to highlight the revisions made (can also seen in the **General Reply**). We are pleased that most reviewers gave our manuscript positive judgements initially, we have strived in the rebuttal to address their remaining concerns and are glad to see this reflected in their feedback.

---

**Reviewer nJkL** assigns a positive score of **6** initally, with remaining suggestions primarily related to implementation details, content organization, and additional experiments on the MITEA dataset and different RAG sources. In response, we provided further clarification of our technical design, reorganized the presentation order of tables to highlight our contribution, and added additional experiments to directly address these points. The reviewer **indicated satisfaction** with our rebuttal.

---

**Reviewer HerW** also assigns a score of **6**. The reviewer mainly suggested 1) adding more discussion with other agentic systems and the Medical World Model; 2) explaining which components in our framework are trained; and 3) addressing implementation details such as the stability of the evidence-based reflection and clinical guideline retrieval. **In response**, we have expanded both the rebuttal and the revised manuscript to include 1) a more detailed discussion contrasting our approach with prior agentic systems and Medical World Model; 2) clarification of our zero-shot design by explicitly stating which components are trained and which are not, and 3) provided additional explanations and supplementary experiments of the robustness of evidence-based reflection and guideline retrieval. We believe the rebuttal have comprehensively addressed the reviewer’s concerns.

---

**Reviewer ACbS** assigns a strong score of **8** and raised questions mainly about technical details, including the composition of the toolbox and the implementation of the evidence-based reflection mechanism. The reviewer also suggested refining the presentation in the main text to make the mathematical notation and the RAG process clearer. Following these suggestions, we expanded the rebuttal and revised manuscript with more detailed technical explanations, simplified and clarified the mathematical definitions, and improved the description of the RAG knowledge base and retrieval process. The reviewer **expressed satisfaction** with our responses and kept the original score unchanged.

---

**Reviewer roDJ** initially assigns a score of **2**, with most comments request for clarification on implementation details (e.g. the implementation of state assessment function or clinical expert evaluation), which are relatively *straightforward to address*. In addition, another concern appear to stem from an interpretation of our work as emphasizing individual module innovation rather than a system-level workflow design. The concern do not relate to fundamental issues of technical soundness or empirical validity.

In our rebuttal, we 1) first reiterated the motivation and significance of our work by highlighting the limitations of existing agentic systems and clarifying how MedAgent-Pro introduces a clinical-aligned agentic framework through system-level design. 2) We then made detailed explanation on all clarification questions raised by the reviewer. Notably, some of these clarifications overlap in rebuttal to other reviewers (*Reviewer ACbS* and *Reviewer nJkL*) and are *indicated effective and convincing* by them.

We have refined the **Introduction** and **Related Work** to more clearly illustrate our contributions, and added detailed explanations in **Section 3.3** and **Appendix A.4, E.3,** and **E.4** to make the technical details clearer and more accessible.  Based on this, we believe that our rebuttal has resolved the majority of Reviewer roDJ’s concerns.

---

We would like to sincerely thank all Area Chairs and reviewers once again for the significant time and effort devoted to the review process. We truly appreciate the careful and constructive feedback, which has helped improve the quality of our paper. We hope that the additional clarifications and analyses provided here will be helpful in supporting a fair and well-informed final decision.

---

### Meta-Review · Area_Chair_STYs · 2026-01-13

**Summary:**

This submission introduces a clinically grounded agentic framework for multimodal medical diagnosis that integrates RAG-based guideline planning, tool-based quantitative analysis, and stepwise reliability checking. The paper is widely viewed as strong, carefully executed, and clinically motivated. The new system-level, clinically grounded AI paradigms has sufficient contributions and novelty for an application, although it seems it integrated multiple know components. The rebuttal is also helpful to clarify few important points.

**Reviewer Concerns:**

three rebuttal moves are helpful:
1. Moving Appendix B.4 into the main paper showed the gains were not just from tool access .
2. turned a vague concept into a rule-based, zero-shot reliability test with explicit criteria (anatomical plausibility, physiological ranges, image quality, etc.) .
3. Demonstrated robustness to guideline source choice, defusing the “weak RAG” criticism .

**Reviewer Scores:**

Only reviewer nJkL could raise his score.

---

### Decision · Program_Chairs · 2026-01-26

Accept (Poster)